# ReCon-GS: Continuum-Preserved Gaussian Streaming for Fast and Compact Reconstruction of Dynamic Scenes

**Jiaye Fu**[1,2], **Qiankun Gao**[1], **Chengxiang Wen**[1], **Yanmin Wu**[1],
**Siwei Ma**[2], **Jiaqi Zhang**[2*], **Jian Zhang**[1,3*]

[1]School of Electronic and Computer Engineering, Peking University
[2]National Engineering Research Center of Visual Technology, Peking University
[3]Guangdong Provincial Key Laboratory of Ultra High Definition Immersive Media Technology,
Peking University

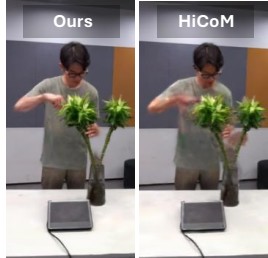 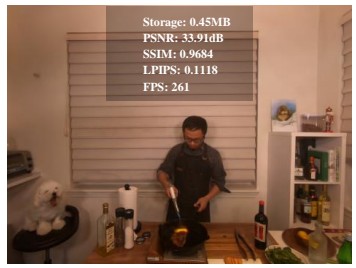 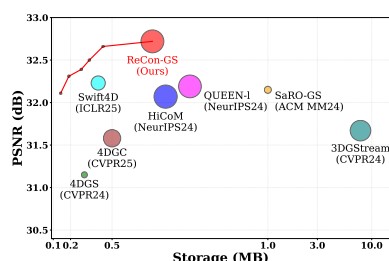

Figure 1: The proposed ReCon-GS framework for streamable dynamic scene reconstruction achieves superior rendering qualty with substantially reduced storage. The left figures show the high-quality rendering results of ReCon-GS in a streaming fashion. The right figure is the performance comparison with previous SOTA [1, 2, 3, 4, 5, 6], where the radius of circle corresponds to the rendering speed.

## Abstract

Online free-viewpoint video (FVV) reconstruction is challenged by slow per-frame optimization, inconsistent motion estimation, and unsustainable storage demands. To address these challenges, we propose the **Re**configurable **Con**tinuum Gaussian Stream, dubbed **ReCon-GS**, a novel storage-aware framework that enables high-fidelity online dynamic scene reconstruction and real-time rendering. Specifically, we dynamically allocate multi-level Anchor Gaussians in a density-adaptive fashion to capture inter-frame geometric deformations, thereby decomposing scene motion into compact coarse-to-fine representations. Then, we design a dynamic hierarchy reconfiguration strategy that preserves localized motion expressiveness through on-demand anchor re-hierarchization, while ensuring temporal consistency through intra-hierarchical deformation inheritance that confines transformation priors to their respective hierarchy levels. Furthermore, we introduce a storage-aware optimization mechanism that flexibly adjusts the density of Anchor Gaussians at different hierarchy levels, enabling a controllable trade-off between reconstruction fidelity and memory usage. Extensive experiments on three widely used datasets demonstrate that, compared to state-of-the-art methods, **ReCon-GS** improves training efficiency by approximately 15% and achieves superior FVV synthesis quality with enhanced robustness and stability. Moreover, at equivalent rendering quality, **ReCon-GS** slashes memory requirements by over 50% compared to leading state-of-the-art methods. Code is avaliable at:
https://github.com/jyfu-vcl/ReCon-GS/.

---

*corresponding to: zhangjian.sz@pku.edu.cn, jqzhang@pku.edu.cn

39th Conference on Neural Information Processing Systems (NeurIPS 2025).

# 1 Introduction

Free-viewpoint Video (FVV) reconstruction has emerged as a cornerstone of next-generation immersive media systems, driven by the escalating demands of XR, AR, and VR applications. While this technology enables dynamic scene exploration from arbitrary viewpoints, its inherent spatiotemporal complexity poses persistent challenges in balancing real-time performance with reconstruction accuracy. Offline approaches [7, 1, 8, 9], though capable of high-quality reconstruction, fail to meet the low-latency requirements of real-time applications such as streaming media and immersive live broadcasting. To address this limitation, recent methods [10, 6] adopt streaming reconstruction frameworks that progressively model inter-frame deformation for frame-by-frame reconstruction.

Despite these advancements, critical bottlenecks persist in achieving practical real-time FVV reconstruction. First, current streaming 3D Gaussian splatting methods incur redundant storage due to deformation fields, primarily attributed to the fundamental incompatibility between the uniform spatial motion encoding strategy of Gaussian primitives and the inherent hierarchical characteristics of physical motion. This mismatch leads to inaccurate and inefficient deformation modeling for dominant macroscopic motions, triggering redundant deformation parameter storage and severe error accumulation issues. While some approaches [11, 12] mitigate error accumulation in streaming motion modeling by introducing auxiliary mechanisms (*e.g.*, optical flow estimation networks), such solutions inevitably increase training costs and significantly compromise practical applicability. Furthermore, Industrial deployment necessitates dynamically balancing storage efficiency and fidelity to accommodate fluctuating resources. However, existing frameworks employ static strategies, thereby lacking the flexibility necessary for practical deployment.

To address the aforementioned challenges, we propose the **Re**configurable **Con**tinuum Gaussian Stream, referred to as **ReCon-GS**, a novel streaming framework that jointly optimizes rendering fidelity and storage adaptability. Our method introduces *Adaptive Hierarchical Motion Representation*, an anchor-driven multi-scale motion encoding paradigm that dynamically allocates Anchor Gaussians through grid-based farthest point sampling to decompose scene dynamics into coarse-to-fine motion structures. This paradigm ensures alignment between Anchor Gaussian distributions and spatial geometric features. Deformation parameters are propagated to clustered General Gaussians via hierarchical mechanisms, enabling effective multi-scale rigid motion representation with minimal storage requirements. Then, **ReCon-GS** implements a *Dynamic Hierarchy Reconfiguration* strategy to resolve the degradation of local motion representation caused by Anchor Gaussian drift during spatial deformation. Specifically, within each hierarchy level, we perform trajectory matching of Anchor Gaussians across frames to track their movement throughout the sequence, while constraining deformation field propagation to operate exclusively within these matched trajectories. Reorganized anchors inherit deformation priors only from their original predecessors at the same hierarchy level, thereby preserving both motion inheritance granularity and temporal motion coherence. Consequently, leveraging these architectural innovations, we reformulate 4D reconstruction as a storage-aware fidelity optimization problem, achieving paradigm-level advancements analogous to rate-distortion optimization theory. This enables dynamic balance between memory efficiency and rendering quality according to application demands without compromising real-time performance, as our framework adaptively adjusts the density of Anchor Gaussians at different hierarchy levels based on scene complexity.

Extensive experiments validate our framework's efficacy across diverse scenarios, demonstrating that **ReCon-GS** achieves compact yet high-fidelity reconstruction under varying motion patterns while outperforming state-of-the-art streaming methods in three well-known datasets. Our key contributions are summarized as follows:

- We propose Adaptive Hierarchical Motion Representation, an anchor-driven multi-scale motion encoding paradigm that adaptively align Anchor Gaussians with scene geometry to decompose scene dynamics into coarse-to-fine structures, achieving an efficient yet compact motion representation.

- We design a Dynamic Hierarchy Reconfiguration strategy to address anchor drift-induced motion degradation, combining re-hierarchization with intra-level deformation inheritance to propagate priors exclusively within original hierarchy layers, eliminating cross-layer interference while preserving motion fidelity.

- Comprehensive experiments across multiple datasets validate **ReCon-GS**'s superiority over state-of-the-art streaming methods. Our framework achieves superior performance with 15%

faster training convergence while reducing storage requirements by over 50% at equivalent rendering quality. Even under constrained storage budgets, **ReCon-GS** maintains significant rendering quality improvements, demonstrating robust practical efficacy.

## 2 Related Work

### 2.1 3D Gaussian Splatting for Static Scenes

With the rise of novel view synthesis (NVS) tasks, 3D Gaussian Splatting (3DGS) [13] has gradually replaced Neural Radiance Fields (NeRF)-based methods [14, 15, 16, 17, 18] as the mainstream approach for static scene reconstruction due to its high rendering quality and fast training speed. However, its massive storage requirements significantly hinder the practical application of 3DGS. Consequently, numerous studies have focused on achieving more compact 3DGS representations while maintaining high-quality rendering. Some methods [19, 20, 21] concentrate on pruning Gaussian primitives, such as removing unimportant Gaussians through trainable mask mechanisms [22] or gradient thresholding approaches [23]. Additionally, other approaches [24] aim to reduce the parameters of 3D Gaussians to decrease per-primitive storage. Meanwhile, 3DGS methods [19, 22, 24, 25, 26] based on Vector Quantization (VQ) principles have also proven effective in reducing storage demands. Alternative strategies [27, 28, 29] explore compact spatial structure representations to minimize 3DGS storage. Scaffold-GS [30] achieves efficient storage reduction by clustering Gaussians to anchor points and implicitly representing Gaussian attributes for each cluster through MLPs. Subsequent works [31, 32, 33] further compress 3DGS by incorporating spatial structure priors into entropy encoders and other compression techniques.

### 2.2 3D Gaussian Splatting of Dynamic Scenes

3D dynamic scene reconstruction stands as one of the most challenging tasks in computer vision and graphics. With the rise of 3D Gaussian Splatting (3DGS), dynamic reconstruction approaches have diverged into two main paradigms. The first involves offline methods [7, 1, 8, 34, 35] that implicitly model scene dynamics through canonical Gaussian fields coupled with spatiotemporal deformation fields. While these methods achieve compact representations, they struggle with efficient motion parameterization. Recent advancements address this by embedding Gaussians into sparse motion representations using anchor points [36, 37, 38] or control points [39, 40, 41], enabling motion propagation through sparse keyframes while preserving reconstruction quality. The second paradigm adopts streaming frameworks that first reconstruct a high-quality 3DGS representation for the initial frame and then model subsequent frames via deformation fields [42, 6, 10]. However, the massive storage overhead of per-frame motion parameters poses significant challenges. Gao et al. [3] mitigate this through a hierarchical motion representation framework, decomposing deformations into coarse-to-fine transformations to reduce parameter dimensionality. Despite this, streaming frameworks inherently suffer from error accumulation due to incremental updates. To address this, some methods [43, 44, 11, 12, 45] leverage pre-trained motion estimation models (e.g., optical flow networks) to guide 3D Gaussian deformations via 2D motion priors, enhancing both rendering quality and temporal consistency. These innovations highlight the critical balance between storage efficiency, computational tractability, and reconstruction fidelity in dynamic 3D modeling.

## 3 Preliminaries

3DGS models scenes using anisotropic 3D Gaussians defined by a mean $\mu$ and a covariance matrix $\mathbf{\Sigma} = \mathbf{R}\mathbf{S}\mathbf{S}^\top\mathbf{R}^\top$, where $\mathbf{S} = diag(s_x, s_y, s_z) \in \mathbb{R}^3$ represents axis-aligned scaling, and $\mathbf{R} \in SO(3)$ is parameterized by a quaternion $\mathbf{q}$. Each Gaussian's view-dependent color uses spherical harmonics $\mathbf{SH}$ and opacity $\alpha \in [0, 1]$.

The mathematical formulation of Gaussian is:

$$\mathcal{G}(\mathbf{x}) = e^{-\frac{1}{2}(\mathbf{x}-\boldsymbol{\mu})^\top\mathbf{\Sigma}^{-1}(\mathbf{x}-\boldsymbol{\mu})}. \tag{1}$$

For novel view synthesis, Gaussians are projected onto the imaging plane using a view transformation matrix $\mathbf{W}$. The projected 2D covariance $\mathbf{\Sigma}'$ is computed through the Jacobian $\mathbf{J}$ of the projective transformation:

$$\mathbf{\Sigma}' = \mathbf{J}\mathbf{W}\mathbf{\Sigma}\mathbf{W}^\top\mathbf{J}^\top. \tag{2}$$

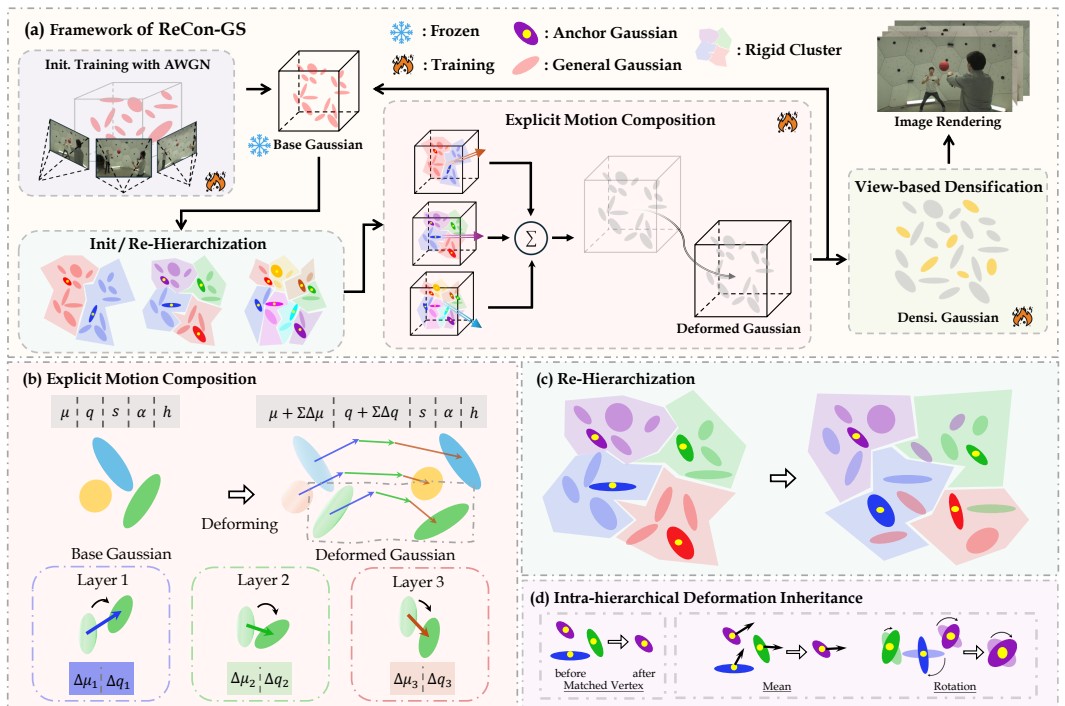

Figure 2: **Illustration of our ReCon-GS framework.** (a) We begin by generating a high-quality base 3D Gaussian Splatting (3DGS) representation for the first frame, then embed it into an adaptively hierarchical motion presentation framework via grid-based farthest-point sampling. (b) Explicit Motion Composition updates the base 3DGS across successive frames. (c) and (d) Periodically, a Re-Hierarchization stage accommodates complex object motion while preserving temporal consistency through Intra-hierarchical Deformation Inheritance. Finally, a view-based densification further refines the 3DGS for high-quality rendering.

Rendering proceeds by alpha-blending $N$ depth-ordered Gaussians at each pixel, with the final color calculated as:

$$C = \sum_{i \in N} \mathbf{c}_i \alpha_i \prod_{j=1}^{i-1} (1 - \alpha_j),$$ (3)

where $\mathbf{c}_i$ and $\alpha_i$ correspond to the color and blending weight of the $i^{th}$ Gaussian, respectively.

The learning pipeline iteratively alternates between gradient-based parameter updates and topological adaptation. Parameter optimization is supervised by the $\mathcal{L}_1$ loss and D-SSIM term:

$$\mathcal{L} = (1 - \lambda)\mathcal{L}_1 + \lambda \mathcal{L}_{\text{D-SSIM}},$$ (4)

where $\lambda$ is typically set to 0.2.

## 4 Methodology

### 4.1 Overview

Our goal is to achieve high-quality dynamic scene reconstruction from multi-view videos under varying storage constraints. To accomplish this, our framework first generates compact Base Gaussians $\mathcal{G}_B$ through an additive noise-injected 3DGS training strategy [3] during initial frame reconstruction. Subsequently, to efficiently parameterize per-frame Gaussian deformations, we propose an adaptive hierarchical motion representation framework that explicitly models scene dynamics through scale-aware motion decomposition. Specifically, each Gaussian's motion is parameterized as a composition of layer-wise rigid transformations across multiple geometric scales, with each deformation layer governed by a distinct set of Anchor Gaussians $\mathcal{G}_A$ spatially distributed via grid-based farthest-point sampling. General Gaussians within each layer hierarchically inherit blended motion parameters from

their nearest anchor counterparts through cross-scale deformation coefficients, enabling progressive motion aggregation from coarse to fine hierarchies.

To enhance the adaptability of our anchor-based motion representation for scenes with varying motion magnitudes, we introduce dynamic hierarchy reconfiguration - a self-optimizing mechanism that redistributes Anchor Gaussians to match evolving scene geometry while maintaining temporal motion consistency through Intra-hierarchical Deformation Inheritance. Finally, to further refine local details, ReCon-GS optimizes 3DGS representation through a view-adaptive densification process, enhancing scene rendering fidelity. The complete pipeline is illustrated in Figure 2 (a).

## 4.2 Adaptively Hierarchical Motion Presentation

Inspired by the hierarchical decomposition of real-world object motion into multi-scale rigid-body components, we propose an adaptive hierarchical motion representation framework. Our framework begins with the **Base Gaussians** $\mathcal{G}_B$, which is the static 3D Gaussians reconstructed from the first frame of multi-view inputs. Building upon $\mathcal{G}_B$, we establish an adaptive hierarchical motion representation through three core stages:

**Anchor Gaussian Initialization**. Given $\mathcal{G}_B = \{g_i\}_{i=1}^N$ with position $\mu_i \in \mathbb{R}^3$, we first compute the scene's spatial bounds as $\mu_{min} = \min_i \mu_i$ and $\mu_{max} = \max_i \mu_i$. $\mathcal{G}_B$ is then divided into a uniform 3D grid with resolution $M = \left\lceil N_{anchor}^{1/3} \right\rceil$, where $N_{anchor}$ is the number of anchor. For each cell $\mathcal{C}_{i,j,k}$, we select the **Anchor Gaussians** $g_a \in \mathcal{G}_A$ with position $\mu_a$ closest to its geometric center:

$$\mathbf{g}_a = \arg \min_{\mathbf{g} \in \mathcal{C}_{i,j,k}} \|\boldsymbol{\mu} - \mathbf{c}_{i,j,k}\|_2, \quad where \quad \mathbf{c}_{i,j,k} = \boldsymbol{\mu}_{\min} + \left(i + \frac{1}{2}, j + \frac{1}{2}, k + \frac{1}{2}\right) \odot \frac{\boldsymbol{\Delta}}{M}, \quad (5)$$

with $\Delta = \mu_{max} - \mu_{min}$. This *grid-based farthest point sampling* ensures each anchor locally represents its cell while maintaining global uniformity. Compared to conventional farthest point sampling approaches, our grid-based strategy achieves significant computational acceleration through structured sampling constraints, effectively reducing the time complexity while preserving sampling quality.

**Hierarchical Rigid-Cluster Formation**. With $\mathcal{G}_A$ initialized, each **General Gaussians** $g_n \in \mathcal{G}_N$ is assigned to its nearest anchor via L1-distance minimization:

$$g_a^* = \arg \min_{g_a \in \mathcal{G}_A} \|\mu_n - \mu_a\|_1. \quad (6)$$

forming primary rigid clusters $\{\mathcal{R}_a\}$. To address residual motions within rigid clusters, we recursively subdivide the hierarchy into two finer levels within the original spatial coordinate system. At each subsequent level, we increase the density of Anchor Gaussians by adapting the grid resolution $M = \left\lceil (N_{anchor} \cdot 3^{l-1})^{1/3} \right\rceil$, where $l = 2, 3$ denotes the hierarchy level.

**Explicit Motion Composition**. As shown in Figure 2 (b), each hierarchy level $l \in \{1, 2, 3\}$ associates clusters with rigid transformations $\mathbf{T}^{(l)} = (\Delta\mu^{(l)}, \Delta\mathbf{q}^{(l)},)$, where $\Delta\mu^{(l)} \in \mathbb{R}^3$ and $\Delta\mathbf{q}^{(l)} \in \mathbb{R}^4$ denote incremental translation and rotation of Anchor Gaussians $\mathcal{G}_A$. Thus, Anchor Gaussians $\mathcal{G}_A$ act as rigid motion keypoints that govern the affine transformation of all General Gaussians $\mathcal{G}_N$ within their associated cluster. By propagating these parameters hierarchically, the motion of each Anchor dictates the collective deformation of its rigid sub-region, emulating real-world rigid-body dynamics. The total deformation of a Gaussian is the cumulative sum across levels:

$$\Delta\boldsymbol{\mu}_h = \sum_{l=1}^3 \Delta\boldsymbol{\mu}^{(l)}, \quad \Delta\boldsymbol{q}_h = \sum_{l=1}^3 \Delta\boldsymbol{q}^{(l)}. \quad (7)$$

Upper level model global rigid motions shared across clusters, while the finest level resolves residual deformations unique to individual Gaussians. Crucially, we decouple geometric attributes (optimized via $\Delta\boldsymbol{\mu}_h$ and $\Delta\boldsymbol{q}_h$) from static appearance parameters (sphere harmonic coefficient $SH$ and opacity $\alpha$), freezing the latter to avoid parameter explosion from separately modeling appearance deformations, which is critical for ReCon-GS's storage efficiency. Since pixel intensities rely on $\alpha$-blended Gaussian contributions, precise geometric deformation alone ensures photorealistic rendering. By optimizing geometry while freezing appearance attributes, ReCon-GS achieves parameter efficiency without sacrificing visual quality.

**Motion Parameter Preservation.** ReCon-GS can store motion parameters at each level more compactly compared to previous streaming reconstruction method [3]. This is primarily due to the use of a carefully designed *grid-based farthest point sampling* algorithm for anchor Gaussian assignment, instead of a uniform anchor sampling strategy. As a result, ReCon-GS can store motion parameters for Anchor Gaussians based solely on their relative positions, without the need for additional indices. This significantly reduces the memory required for storing motion parameters.

## 4.3 Dynamic Hierarchy Reconfiguration

Using Adaptive Hierarchical Motion Representation framework, we efficiently allocates explicit motion parameters according to scene density and geometric structures. However, ReCon-GS also inherits a critical challenge common to streaming reconstruction methods [6, 3]: persistent scene motion induces severe deformations in local rigid clusters, causing Anchor Gaussians to gradually lose their representativeness as rigid motion bases. This degradation propagates errors through the motion hierarchy, accumulating artifacts in rendered sequences. To address this, we propose Dynamic Hierarchy Reconfiguration strategy—a self-correcting mechanism that periodically reinitializes the hierarchical structure while preserving temporal motion coherence through intra-hierarchical deformation inheritance.

**Periodic Re-Hierarchization**. At each reconfiguration step $t = kT$ ($k \in \mathbb{N}^+$), the framework regenerate Anchor Gaussians $\mathcal{G}_A$ using grid-based farthest point sampling on the current Gaussian distribution, ensuring updated anchors align with the latest scene geometry. Then, General Gaussian $\mathcal{G}_N$ is reassigned to new anchors via L1-distance minimization. The process is described in Figure 2 (c).

**Intra-hierarchical Deformation Inheritance**. As present in Figure 2 (d), after re-hierarchization, we implement intra-hierarchical deformation inheritance to enforce temporal motion consistency between $\mathcal{G}_A$ across hierarchy levels. For each hierarchy level $l \in \{1, 2, 3\}$, each new Anchor Gaussian inherits deformation parameters from its three nearest legacy Anchors to ensure temporal continuity. Specifically, The translational deformation is governed by:

$$\Delta \boldsymbol{\mu}_a'^{(l)} = \frac{1}{3} \sum_{i=1}^{3} \Delta \boldsymbol{\mu}_{a_i}^{(l)},\tag{8}$$

where $\Delta \boldsymbol{\mu}_a'^{(l)}$ denotes the translational deformation parameter of new Anchor Gaussian at level $l$, and $\Delta \boldsymbol{\mu}_{a_i}^{(l)}$ represents the translational deformation parameter of the $i$-th matched Anchor Gaussian. Rotational deformation follows an eigenvector-based formulation:

$$\Delta \boldsymbol{q}_a'^{(l)} = \frac{v_{\max}(\mathcal{M})}{\left\| v_{\max}(\mathcal{M}) \right\|}, \quad \mathcal{M} = \sum_{i=1}^{3} \Delta \boldsymbol{q}_{a_i}^{(l)} \big(\Delta \boldsymbol{q}_{a_i}^{(l)}\big)^{\mathsf{T}},\tag{9}$$

where $\Delta \boldsymbol{q}_a'^{(l)}$ is the rotational deformation parameter of new Anchor Gaussian, and $\Delta \boldsymbol{q}_{a_i}^{(l)}$ represents the rotational deformation parameter of the $i$-th matched Anchor Gaussian. $v_{max}(M)$ denotes the eigenvector corresponding to the largest eigenvalue of $\mathcal{M}$.

## 4.4 Storage-aware Optimization

Thanks to the streaming architecture of ReCon-GS, the storage overhead of Gaussian primitives constitutes only a minor fraction of the total storage consumption, with the majority allocated to per-frame deformation field parameters. By adopting an anchor-driven adaptive hierarchical motion representation, ReCon-GS operates as a storage-aware framework—the hierarchical anchor resolution $\{N_{anchor}^{(l)}\}_{l=1}^{3}$ dynamically adapts to varying scene complexity and storage constraints. Moreover, ReCon-GS adopts a decoupled two-stage optimization pipeline to balance motion fidelity and computational efficiency. In Phase 1, we exclusively train the hierarchical deformation fields while freezing the geometric and appearance attributes of the Base Gaussians $\mathcal{G}_B$. Once the deformation fields converge, Phase 2 activates the view-based densification to address persistent discrepancies between learned motion and scene dynamics.

Table 1: **Quantitative comparison** on N3DV dataset. The storage metric includes the size without and with the initial frame, separated by "/". The training time metric includes the first frame training. The method with † is reproduced by us through official code in the same experimental environment.

| Category | Method | PSNR (dB)↑ | SSIM↑ | LPIPS↓ | Storage (MB)↓ | Train (sec)↓ | Render (FPS)↑ |
|---|---|---|---|---|---|---|---|
| Offline | 4DGS† [1] | 31.36 | 0.950 | 0.131 | 0.3 | 7.8 | 30 |
| | STG [9] | 32.05 | 0.948 | - | 0.67 | 20 | 140 |
| | SaRO-GS [5] | 32.15 | - | - | 1.0 | - | 40 |
| | Swift4D [46] | 32.23 | - | - | 0.4 | 5.0 | 125 |
| | SplineGS [47] | 32.60 | - | - | - | 11 | 76 |
| Online | Dynamic 3DGS [42] | 30.67 | - | - | -/9.2 | 560 | - |
| | StreamRF [48] | 30.68 | 0.930 | - | 17.7/31.5 | 15 | 12 |
| | 3DGStream† [6] | 31.35 | 0.948 | 0.130 | 7.6/7.8 | 8.1 | 245 |
| | 4DGC [2] | 31.58 | 0.943 | - | -/0.5 | 50 | 168 |
| | QUEEN-l [4] | 32.19 | 0.946 | 0.136 | -/0.75 | 7.9 | 248 |
| | HiCoM† [3] | 32.08 | 0.953 | 0.130 | 0.48/0.69 | 6.6 | 255 |
| | **ReCon-GS** (ours) | 32.66 | 0.957 | 0.123 | 0.40/0.44 | 6.4 | 250 |

Table 2: **Quantitative comparison** on Meet Room and PanopticSports datasets. The storage metric includes the size without and with the initial frame, separated by "/". The training time metric includes the first frame training. The method with † is reproduced by us through official code in the same experimental environment. The method marked with * adopts the *Discussion* scene from the Meet Room dataset as the training set, with the remaining scenes serving as the test set.

| Method | Meet Room | | | | | PanopticSports | | | | |
|---|---|---|---|---|---|---|---|---|---|---|
| | PSNR (dB)↑ | LPIPS↓ | Storage (MB)↓ | Train (sec)↓ | Render (FPS)↑ | PSNR (dB)↑ | LPIPS↓ | Storage (MB)↓ | Train (sec)↓ | Render (FPS)↑ |
| 3DGStream† [6] | 29.30 | 0.188 | 4.0/4.1 | 4.77 | 260 | 23.02 | 0.187 | 7.9/8.1 | 5.87 | 369 |
| IGS-l* [12] | 30.13 | - | -/1.26 | 2.67 | 252 | - | - | - | - | - |
| HiCoM† [3] | 29.57 | 0.182 | 0.30/0.39 | 3.91 | 236 | 29.17 | 0.142 | 1.33/2.11 | 8.60 | 358 |
| **ReCon-GS** (ours) | 30.84 | 0.163 | 0.28/0.30 | 3.86 | 256 | 29.33 | 0.136 | 0.64/0.8 | 7.14 | 410 |

## 5 Experiments

### 5.1 Dataset

**The Nerual 3D Video Dataset (N3DV)** [51] comprises 6 indoor dynamic scenes. Each scene contains 18 to 21 free-viewpoint videos, with each video spanning 300 frames captured at 30 FPS and a resolution of $2704 \times 2028$.

**Meeting Room Dataset** [48] comprises 3 dynamic scenes across diverse real-world scenarios. Each scene contains 13 free-viewpoint videos, with each video spanning 300 frames captured at 30 FPS and a resolution of $1280 \times 720$.

**PanopticSports Dataset** [42], derived from sports sequences in the Panoptic Studio dataset [52], includes 6 sports-oriented scenarios. Each scenario consists of 31 free-viewpoint videos, each containing 150 frames recorded at 30 FPS and a resolution of $640 \times 360$.

**Technicolor Dataset** [53] consists of video recordings captured by a $4 \times 4$ camera rigs. Each video has a spatial resolution of $2048 \times 1088$ and a frame rate of 30 FPS. Following the Ex4DGS [34], evaluation is performed on five distinct scenes (*Birthday, Fabien, Painter, Theater, and Trains*) using their original full resolution.

### 5.2 Implementation Details

Our framework initializes first-frame reconstruction using 3DGS [13] with the maximum degree of spherical harmonics (SH) to 1 and a noise-injected coefficient $\lambda_{noise} = 0.01$, training on three datasets for 10,000 to 15,000 steps. During the construction of hierarchical motion representation,

Table 3: **Quantitative comparison** on Technicolor dataset. The storage metric includes the size with the initial frame. The training time metric includes the first frame training.

| Category | Method | PSNR (dB)↑ | SSIM↑ | Storage (MB)↓ | Render (FPS)↑ |
|----------|--------|-----------|-------|---------------|---------------|
| NeRF-based | HyperReel [49] | 31.80 | 0.906 | 1.20 | 4 |
| Offline | STG [9] | 33.60 | - | 1.10 | 87 |
| | Ex4DGS [34] | 33.62 | 0.916 | 2.81 | 72 |
| Online | E-D3DGS [50] | 33.24 | 0.907 | 1.54 | 79 |
| | **ReCon-GS** (ours) | 33.83 | 0.932 | 0.82 | 207 |

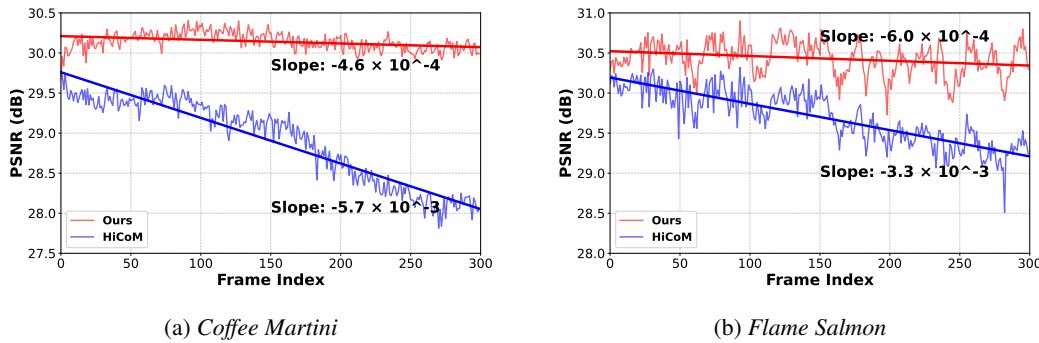

(a) *Coffee Martini*           (b) *Flame Salmon*

Figure 3: The PSNR Trend Comparison between Ours and HiCoM [3] across different scenarios.

we set the finest-level Anchor Gaussians to 1/24 of the total Gaussian primitive count, with each subsequent layer containing one-third the anchors of its predecessor.

For per-frame optimization, all datasets undergo an initial 100 steps for deformation training. Subsequently, N3DV and Meet Room datasets receive an additional 100 refinement steps for Gaussian densification and pruning, while PanopticSports employs 400 steps to address its higher motion complexity. All experiments are conducted on an NVIDIA RTX 4090 GPU, with results averaged over 3 independent runs to ensure statistical reliability. Extended implementation details are provided in the **Appendix A**.

### 5.3 Experiment Results

**Quantitive comparisons.** We conducted a comprehensively evaluation of our method against existing state-of-the-art (SOTA) online and offline approaches across three datasets using multiple quantitative metrics, including PSNR, SSIM, LPIPS, storage consumption, training time, and rendering speed. To ensure statistical validity, storage and rendering speed measurements were calculated as averaged values across all video frames. As demonstrated in Table 1 , Table 2 and Table 3, our method achieves SOTA performance in rendering quality and training efficiency compared to online baselines. Specifically, our method outperforms the recent SOTA streaming methods in rendering quality by **more than 0.5dB** on the N3DV [51], MeetRoom [48], and Technicolor [53] datasets. This improvement underscores the effectiveness of our approach in achieving superior visual fidelity in dynamic scene reconstructions. In the case of the PanopticSports [42] dataset, our method achieves the best rendering quality while inducing a substantial reduction in storage requirements, decreasing storage by approximately 60% compared to previous streaming methods. In comparison to offline methods, ReCon-GS requires slightly more storage, primarily due to the inherent storage advantages of offline methods in implicit motion representation.

Furthermore, as evidenced in Figure 3, ReCon-GS effectively alleviates temporal performance degradation observed in recent SOTA method. Through the Dynamic Hierarchy Reconfiguration Strategy, ReCon-GS successfully circumvents the degradation in motion representation capability of Anchor Gaussians. More quantitive evaluations are provided in the **Appendix C.1**.

**Qualitative comparisons.** While our ReCon-GS framework primarily focuses on optimizing the efficiency of dynamic scene reconstruction, Table 1 and Table 2 substantiate the superiority of our

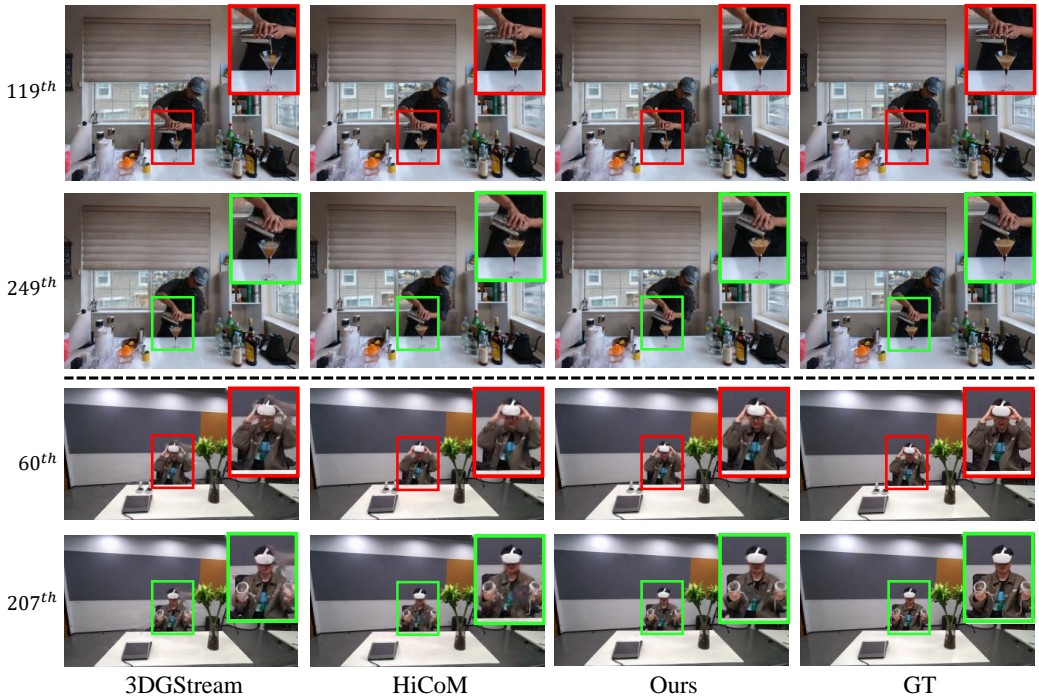

Figure 4: **Qualitative results** on the *coffee martini* scene in the N3DV Dataset. To ensure a fair comparison, we retrained their official code with the same initial sparse points.

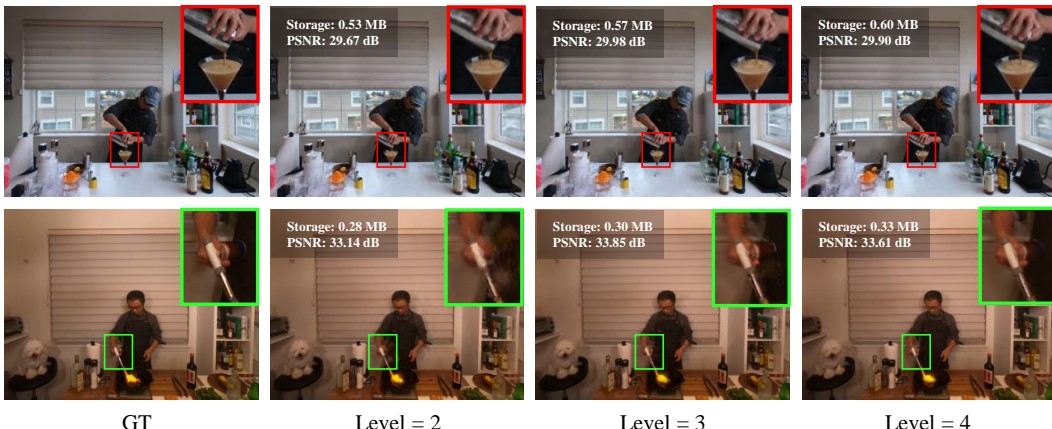

Figure 5: Qualitative results of our ReCon-GS under different hierarchical levels

method in rendering quality. As illustrated in Figure 4, compared to HiCoM [3] and 3DGStream [6], our approach achieves more complete and temporally consistent reconstruction of dynamic scene details across multiple datasets. Moreover, the proposed method demonstrates more enhanced dynamic reconstruction capabilities when handling prolonged temporal intervals. We have also provided an additional subjective quality evaluations in **Appendix C.3**.

## 5.4 Ablation Study

As evidenced in Table 4, we conducted ablation studies on key strategies of ReCon-GS, including Adaptively Hierarchical Motion Presentation paradigm, Dynamic Hierarchy Reconfiguration strategy and view-based densification process. The experimental results demonstrate that the Adaptively Hierarchical Motion Presentation paradigm, despite introducing some memory overhead, significantly enhances the performance of ReCon-GS, proving the paradigm's effective motion representation

Table 4: **Ablation on key components of our ReCon-GS framework**.

| Method | PSNR (dB)↑ | SSIM ↑ | Storage (MB)↓ | Train (sec)↓ | Render (FPS) ↑ |
|---|---|---|---|---|---|
| w/o Hierarchical Motion Representation | 31.43 | 0.9525 | **0.32** | **6.37** | 260 |
| w/o Dynamic Hierarchy Reconfiguration | 32.00 | 0.9521 | 0.45 | 6.45 | 248 |
| w/o View-based Densification | 32.15 | 0.9559 | 0.40 | 6.52 | **287** |
| Ours (full) | **32.66** | **0.9571** | 0.44 | 6.44 | 250 |

Table 5: **Ablation on hierarchy depth of our ReCon-GS framework**.

| Hierarchy Depth | N3DV | | | Meet Room | | |
|---|---|---|---|---|---|---|
| | PSNR (dB) ↑ | Storage (MB) ↓ | Train (sec) ↓ | PSNR (dB) ↑ | Storage (MB) ↓ | Train (sec) ↓ |
| 2 | 32.422 | **0.40** | **6.44** | 30.20 | **0.28** | 3.87 |
| 3 | **32.662** | 0.44 | 6.49 | 30.84 | 0.30 | **3.86** |
| 4 | 32.658 | 0.46 | 6.54 | **31.01** | 0.32 | 4.01 |

capabilities. Furthermore, the implementation of Dynamic Hierarchy Reconfiguration strategy further improves ReCon-GS's rendering quality, indicating enhanced spatiotemporal localization accuracy of Gaussian primitives and ensured temporal motion consistency. While the view-based densification process increases per-frame training time, it substantially improves the representation of high-frequency scene details. This suggests that merely deforming base Gaussian primitives is insufficient for comprehensive dynamic scene modeling.

Furthermore, Table 5 presents ablation experiments on hierarchy depth, motivated by our investigation into motion representation efficiency through ReCon-GS. As shown in Figure 5, experimental findings reveal that a 3-level hierarchy optimally balances high-fidelity scene reconstruction with storage efficiency. This configuration enables ReCon-GS to achieve superior fidelity-storage trade-offs compared to other hierarchical configurations. See **Appendix B** for more ablation studies.

## 6 Conclusion

This paper introduces **ReCon-GS**, a novel storage-aware framework for high-fidelity multi-view video reconstruction. ReCon-GS achieves density-aware multi-layer decoupled compact motion modeling of Gaussian primitives through an Adaptive Hierarchical Motion Representation framework. Furthermore, we propose a Dual-level Deformation Strategy combining inter-frame temporal deformation and intra-frame spatial deformation, accurately capture spatio-temporal positions of Gaussian primitives while avoiding reconstruction failures in high-dynamic motions and emerging objects. Notably, ReCon-GS innovatively transforms dynamic scene reconstruction from conventional single-objective optimization into a storage-aware fidelity optimization framework. This paradigm shift allows dynamic balancing between memory efficiency and rendering quality according to application requirements, significantly enhancing the practical applicability of 3D dynamic reconstruction method. Extensive experiments demonstrate that ReCon-GS achieves state-of-the-art performance in both rendering quality and training efficiency while maintaining exceptional storage effectiveness. Compared with existing SOTA methods, ReCon-GS attains over 50% memory reduction while preserving same rendering fidelity.

**Limitations.** While maintaining alignment with prior methodologies in focusing on multi-view general scene reconstruction tasks, our approach is not specifically optimized for monocular datasets or specialized reconstruction scenarios. Similar to existing streaming-based architectures, we face challenges in dynamic allocation/updating of Gaussians to handle real-time object emergence/dissolution, as abrupt scene changes require instant 3DGS structural adaptations.

## 7 Acknowledge

This work was supported in part by the Guangdong Provincial Key Laboratory of Ultra High Definition Immersive Media Technology (No. 2024B1212010006) and Beijing Natural Science Foundation under grant no. L242014, in part by the PCL-CMCC Foundation for science and innovation grant 2024ZY1C0040, which are gratefully acknowledged.

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

# Appendix

## A  More Implementation Details

We implement ReCon-GS upon the open-source codebase of 3D Gaussian Splatting (3DGS) [13]. During initial frame training, we set the spherical harmonic coefficients (SH) to degree 1 to ensure the storage compactness. For all datasets, we terminate the densification process of 3DGS at 5,000 iterations, which significantly alleviates the overfitting issue in streaming 3DGS during initial frame training. Additionally, we adopt the opacity preservation strategy from 4DGS to effectively eliminate oversized Gaussian primitives. The hyperparameter $\lambda_{noise}$ follows the first-frame configuration from HiCoM [3], while other 3DGS parameters maintain their default settings in 3DGS. Specifically, we train the first frame for 10,000 iterations on the Meet Room dataset, and extend this to 15,000 iterations for both N3DV and PanopticSports datasets.

For subsequent frame optimization, our approach differs from HiCoM's implementation which uses 250 iterations for deformation field training on Panoptic Sports versus 100 iterations on N3DV and Meet Room datasets. We uniformly apply 100 iterations for deformation field training across all datasets due to our superior ability in capturing the motion of Gaussian primitives. In Phase 2, the refinement of deformed Gaussian primitives, considering the more pronounced surface deformation in PanopticSports dataset compared to other datasets, we implement 400 iterations for intensive refinement on PanopticSports dataset versus 100 iterations for N3DV and MeetRoom datasets. All 3DGS-related hyperparameters remain consistent with those used in initial frame training to ensure methodological coherence.

## B  More Ablation Studies

As ReCon-GS represents a novel storage-aware framework designed for high-fidelity real-time rendering in dynamic scene reconstruction, the quantity of Anchor Gaussians per layer in our adaptive hierarchical motion representation structure fundamentally governs the critical trade-off between storage efficiency and rendering quality. Concurrently, the iteration counts allocated to each training stage in subsequent frame training directly determine the training time of our optimization process.

Table 6 investigates the impacts of different iteration configurations across training phases on ReCon-GS performance. The results indicate that prolonging the deformation field training process in Phase 1 can effectively improve rendering quality, however, this training configuration significantly increases computational time. The view-based densification process in Phase 2 demonstrates optimal performance and best storage quality at 100 steps. As evidenced by Table 4 and Table 6, view-based densification effectively address persistent discrepancies between learned motion and scene dynamics, though their impact on overall rendering quality remains less substantial compared to the training of adaptively hierarchical deformation.

We also performed an ablation study on the performance of ReCon-GS, varying the training duration of the initial frame (Frame 0). The detailed results is shown in Table 7. At 2,500 iterations, the suboptimal quality and sparse representation of the initial reconstruction lead to the steepest performance degradation slope. Increasing the training to 5,000 iterations substantially mitigates this quality degradation. Extending the training to 10,000 iterations yields a converged initial frame, which further minimizes quality decay while achieving state-of-the-art performance. Importantly, with the converged initial frame, ReCon-GS requires only a minimal number of incremental Gaussians per frame to ensure high-quality reconstruction. This result strongly validates the superior robustness of

Table 6: **Ablation on training steps of our ReCon-GS framework**.

| Phase 1 | Phase 2 | PSNR (dB)↑ | SSIM ↑ | Storage (MB)↓ | Train (sec)↓ | Rendering (FPS) ↑ |
|---|---|---|---|---|---|---|
| 50 | 100 | 32.40 | 0.9560 | **0.44** | **5.20** | 248 |
| 150 | | **32.76** | 0.9568 | **0.44** | 7.66 | 249 |
| 100 | 50 | 32.56 | 0.9564 | 0.46 | 5.22 | 246 |
| | 150 | 32.64 | 0.9568 | 0.49 | 7.70 | 249 |
| 100 | 100 | 32.66 | **0.9571** | **0.44** | 6.44 | **250** |

Table 7: **Ablation on the Training Steps of Frame 0**. The Init. 3DG refers to the initial 3D Gaussians constructed from the Frame 0. The Incr. 3DG denotes the incremental Gaussians added in Phase 2 for all following frames.

| Training Steps | PSNR (dB) ↑ | Frame 0 PSNR (dB) ↑ | Slope ↓ | Storge (MB) ↓ | Init. 3DG Storage (MB) ↓ | Incr. 3DG Storage (MB) ↓ |
|---|---|---|---|---|---|---|
| 2500 | 31.03 | 31.59 | 0.0077 | 0.46 | 8.96 | 42.17 |
| 5000 | 32.24 | 32.51 | 0.0029 | 0.48 | 11.14 | 11.57 |
| 10000 | 32.63 | 32.75 | 0.0016 | 0.46 | 11.12 | 4.11 |

Table 8: **First-frame reconstruction results across various scenario configurations.** 3DGStream uses standard 3DGS training, Our Method and HiCoM use standard 3DGS training that incorporates with positional noise injection.

| Method | Coffee Martini | | | | Flame Steak | | | | Sear Steak | | | |
|---|---|---|---|---|---|---|---|---|---|---|---|---|
| | PSNR (dB)↑ | GS Num (k)↓ | Storage (MB)↓ | Train (Sec)↓ | PSNR (dB)↑ | GS Num (k)↓ | Storage (MB)↓ | Train (Sec)↓ | PSNR (dB)↑ | GS Num (k)↓ | Storage (MB)↓ | Train (Sec)↓ |
| 3DGStream | 27.62 | 416 | 7.60 | 340 | 33.85 | 208 | 7.60 | 304 | 29.25 | 214 | 7.60 | 317 |
| HiCoM | 29.73 | 348 | 83.00 | 368 | 34.27 | 187 | 44.55 | 334 | 33.96 | 191 | 45.60 | 317 |
| **Ours** | 29.67 | 353 | 16.17 | 354 | 31.20 | 189 | 8.65 | 324 | 33.90 | 191 | 8.76 | 338 |

ReCon-GS against error accumulation and underscores the critical impact of initial frame quality in streaming frameworks.

# C More Detailed Results

## C.1 Quantitative Results

### C.1.1 Initial 3DGS Comparison

For streaming 3DGS reconstruction frameworks, the initial frame training plays a pivotal role in determining the reconstruction quality of subsequent frames. Table 8 presents comparative metrics of average initial frame training quality and corresponding storage consumption across three benchmark datasets. While both ReCon-GS and HiCoM implement positional noise injection during initial Gaussian optimization – a strategic commonality that differentiates them from 3DGStream – our framework demonstrates superior rendering fidelity. This configuration, combined with our first-frame spherical harmonic (SH) coefficient restriction to degree 1, enables ReCon-GS to achieve more compact Gaussian representations compared to HiCoM's SH degree 3 implementation, significantly enhancing storage efficiency without compromising visual quality.

### C.1.2 Per-Scene Results

Table 9, Table 10, and Table 11 present comprehensive comparative results between ReCon-GS and recent state-of-the-art methods across three benchmark datasets under specific scene.

### C.1.3 More Results

Figure 6 demonstrates the superior rate-distortion performance of our method compared to existing SOTA approaches across multiple scenarios. Table 12 presents the detailed results under different hierarchical anchor densities. By modulating hierarchical anchor density $\{N_{anchor}^{(l)}\}_{l=1}^{3}$ and leveraging the high-efficiency characteristics of the Adaptively Hierarchical Motion Presentation paradigm, ReCon-GS maintains robust rendering quality under varying storage constraints.

## C.2 Anchor Gaussian Visualization

We visualize the distribution of Anchor Gaussians at different levels for ReCon-GS and HiCoM, with the results shown in Figure 7. As shown in the Figure 7, compared to the baseline method HiCoM, ReCon-GS assigns more Anchor Gaussians to motion-rich and texture-rich regions (*e.g.*, people, VR glasses, flowers). This is primarily due to the density-based Anchor Gaussian assignment mechanism used by ReCon-GS. As a result, ReCon-GS is able to perform more refined rigid motion representation in areas with rich textures, thus improving the overall rendering quality.

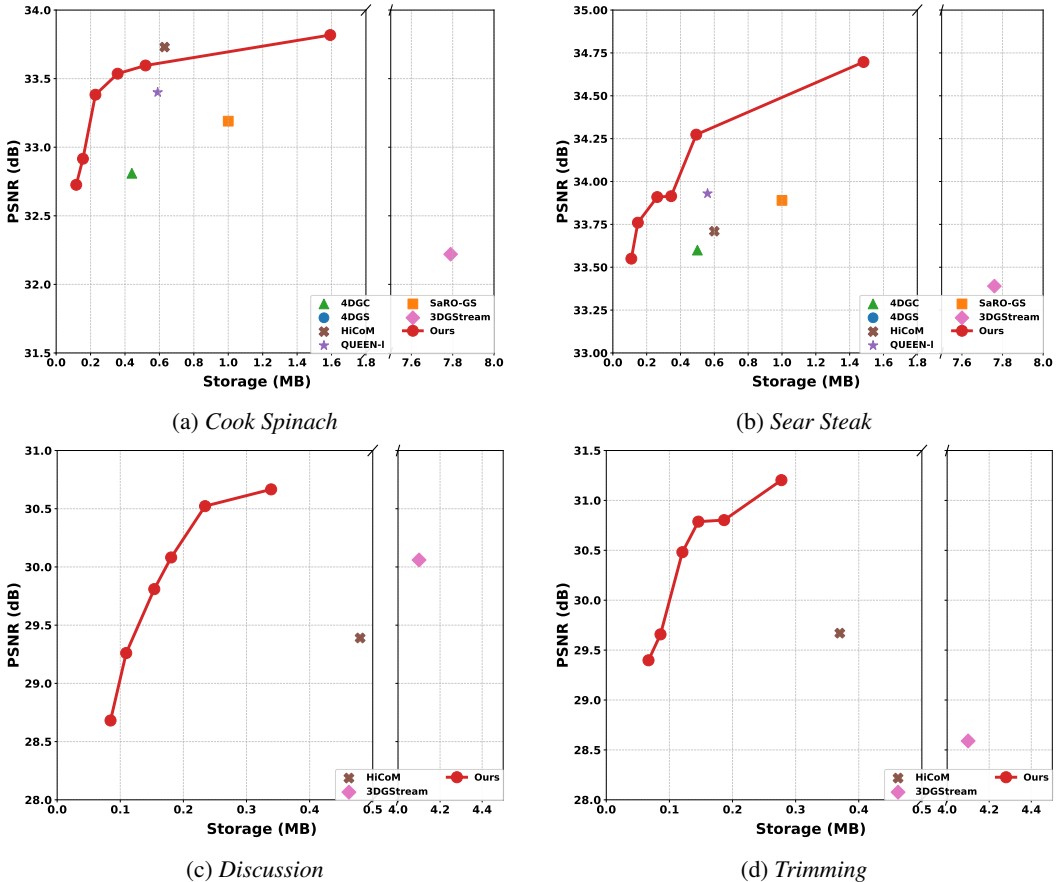

(a) *Cook Spinach*  (b) *Sear Steak*

(c) *Discussion*  (d) *Trimming*

Figure 6: **Rate-distortion performance across diverse scenarios.** As evidenced by comparative evaluations, our framework not only achieves superior compression efficiency compared to state-of-the-art approaches [1, 6, 3, 4, 5, 2], but also demonstrates unique adaptability in storage-aware fidelity optimization - a critical capability for practical deployment scenarios where resource constraints and reconstruction quality must be jointly optimized.

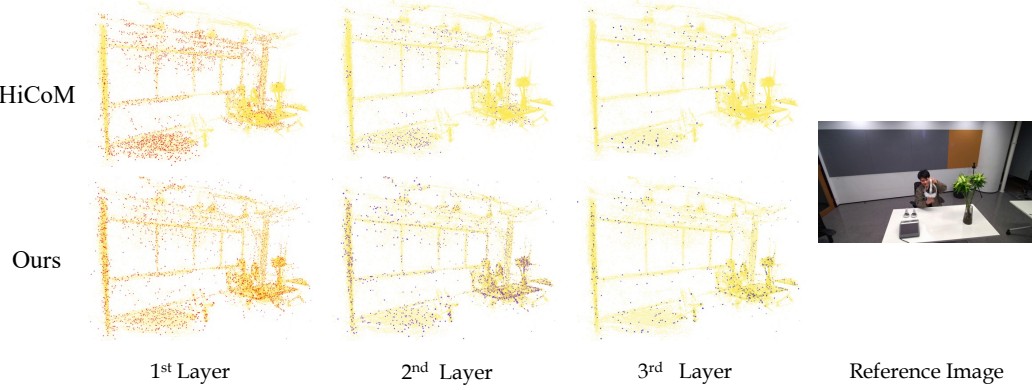

Figure 7: **The Visualization of Anchor Gaussian Distribution.** $1^{st}$ Layer is the finest layer, and $3^{rd}$ Layer is the coarsest layer. The yellow points represent the General Gaussian, while the red, purple, and blue points represent the Anchor Gaussians for their respective layers.

Table 9: **Per-scene quantitative results on the N3DV dataset.** The method with $^\dagger$ is reproduced by us through official code in the same experimental environment.

| Category | Method | Coffee Martini | | | Cook Spinach | | | Cut Roasted Beef | | |
|---|---|---|---|---|---|---|---|---|---|---|
| | | PSNR↑ | SSIM↑ | Storage↓ | PSNR↑ | SSIM↑ | Storage↓ | PSNR↑ | SSIM↑ | Storage↓ |
| Offline | 4DGS$^\dagger$ [1] | 30.10 | 0.935 | 0.30 | 31.15 | 0.958 | 0.30 | 32.60 | 0.951 | 0.30 |
| | SaRO-GS [5] | 28.96 | - | 1.0 | 33.19 | - | 1.0 | 33.91 | - | 1.0 |
| Online | 3DGStream$^\dagger$ [6] | 27.75 | 0.917 | 7.85 | 32.22 | 0.957 | 7.79 | 32.67 | 0.957 | 7.76 |
| | 4DGC [2] | 27.98 | - | 0.58 | 32.81 | - | 0.44 | 33.03 | - | 0.47 |
| | QUEEN-l [4] | 28.38 | 0.915 | 1.17 | 33.40 | 0.956 | 0.59 | 34.01 | 0.959 | 0.57 |
| | HiCoM$^\dagger$ [3] | 28.67 | 0.925 | 0.82 | 33.73 | 0.962 | 0.63 | 32.73 | 0.963 | 0.60 |
| | **Ours** | 30.14 | 0.938 | 0.64 | 33.54 | 0.961 | 0.35 | 33.92 | 0.966 | 0.36 |

| Category | Method | Flame Salmon | | | Flame Steak | | | Sear Steak | | |
|---|---|---|---|---|---|---|---|---|---|---|
| | | PSNR↑ | SSIM↑ | Storage↓ | PSNR↑ | SSIM↑ | Storage↓ | PSNR↑ | SSIM↑ | Storage↓ |
| Offline | 4DGS$^\dagger$ [1] | 30.21 | 0.934 | 0.30 | 33.49 | 0.961 | 0.30 | 30.64 | 0.961 | 0.30 |
| | SaRO-GS [5] | 29.14 | - | 1.0 | 33.83 | - | 1.0 | 33.89 | - | 1.0 |
| Online | 3DGStream$^\dagger$ [6] | 28.61 | 0.924 | 7.83 | 33.47 | 0.966 | 7.79 | 33.39 | 0.965 | 7.76 |
| | 4DGC [2] | 28.49 | - | 0.51 | 33.58 | - | 0.44 | 33.60 | - | 0.50 |
| | QUEEN-l [4] | 29.25 | 0.923 | 1.00 | 34.17 | 0.962 | 0.59 | 33.93 | 0.962 | 0.56 |
| | HiCoM$^\dagger$ [3] | 29.70 | 0.932 | 0.81 | 33.92 | 0.969 | 0.60 | 33.71 | 0.968 | 0.60 |
| | **Ours** | 30.43 | 0.938 | 0.61 | 34.00 | 0.969 | 0.33 | 33.91 | 0.966 | 0.35 |

Table 10: **Per-scene quantitative results on the Meet Room dataset.** The method with $^\dagger$ is reproduced by us through official code in the same experimental environment.

| Category | Method | Discussion | | | Trimming | | | Vrheadset | | |
|---|---|---|---|---|---|---|---|---|---|---|
| | | PSNR↑ | SSIM↑ | Storage↓ | PSNR↑ | SSIM↑ | Storage↓ | PSNR↑ | SSIM↑ | Storage↓ |
| Online | 3DGStream$^\dagger$ [6] | 30.06 | 0.945 | 4.10 | 28.59 | 0.939 | 4.10 | 29.25 | 0.941 | 4.10 |
| | HiCoM$^\dagger$ [3] | 29.39 | 0.941 | 0.48 | 29.67 | 0.945 | 0.37 | 29.65 | 0.947 | 0.30 |
| | **Ours** | 30.67 | 0.955 | 0.35 | 31.20 | 0.956 | 0.28 | 30.64 | 0.953 | 0.30 |

Table 11: **Per-scene quantitative results on the PanopticSports dataset.** The method with $^\dagger$ is reproduced by us through official code in the same experimental environment.

| Category | Method | Basketball | | | Boxes | | | Football | | |
|---|---|---|---|---|---|---|---|---|---|---|
| | | PSNR↑ | SSIM↑ | Storage↓ | PSNR↑ | SSIM↑ | Storage↓ | PSNR↑ | SSIM↑ | Storage↓ |
| Offline | 4DGS$^\dagger$ [1] | 26.99 | 0.912 | 0.34 | 27.76 | 0.922 | 0.32 | 27.20 | 0.922 | 0.36 |
| Online | Dynamic3DGS [42] | 28.22 | 0.910 | 0.74 | 29.46 | 0.910 | 0.74 | 28.49 | 0.910 | 0.74 |
| | 3DGStream$^\dagger$ [6] | 20.08 | 0.815 | 8.10 | 23.76 | 0.861 | 8.10 | 22.80 | 0.856 | 8.10 |
| | HiCoM$^\dagger$ [3] | 28.49 | 0.915 | 2.06 | 29.03 | 0.922 | 2.05 | 29.64 | 0.926 | 2.01 |
| | **Ours** | 28.78 | 0.921 | 0.81 | 28.93 | 0.926 | 0.80 | 30.01 | 0.931 | 0.77 |

| Category | Method | Juggle | | | Softball | | | Tennis | | |
|---|---|---|---|---|---|---|---|---|---|---|
| | | PSNR↑ | SSIM↑ | Storage↓ | PSNR↑ | SSIM↑ | Storage↓ | PSNR↑ | SSIM↑ | Storage↓ |
| Offline | 4DGS$^\dagger$ [1] | 26.72 | 0.923 | 0.32 | 26.88 | 0.923 | 0.31 | 27.64 | 0.925 | 0.31 |
| Online | Dynamic3DGS [42] | 29.48 | 0.920 | 0.74 | 28.43 | 0.910 | 0.74 | 28.11 | 0.910 | 0.74 |
| | 3DGStream$^\dagger$ [6] | 26.25 | 0.891 | 8.10 | 22.22 | 0.858 | 8.10 | 22.99 | 9.862 | 8.10 |
| | HiCoM$^\dagger$ [3] | 29.28 | 0.932 | 2.02 | 29.61 | 0.928 | 1.97 | 28.97 | 0.925 | 2.05 |
| | **Ours** | 29.71 | 0.938 | 0.79 | 29.61 | 0.933 | 0.77 | 28.97 | 0.929 | 0.79 |

Table 12: Quantitative Results under different hierarchical anchor density $\{N_{anchor}^{(l)}\}_{l=1}^{3}$.

| | Hierarchical Anchor Density | | | | | |
|---|---|---|---|---|---|---|
| | 1/16 | 1/24 | 1/32 | 1/48 | 1/64 | 1/96 |
| Storage (MB/Frame) | 0.63 | 0.44 | 0.37 | 0.28 | 0.19 | 0.14 |
| PSNR (dB) | 32.72 | 32.66 | 32.50 | 32.39 | 32.31 | 32.11 |

### C.3 Qualitative Results

Figure 8 provides expanded quantitative comparisons between ReCon-GS and HiCoM on the N3DV dataset, highlighting critical performance differentials in dynamic scene reconstruction.

Figure 9 presents extended visual comparisons on the MeetRoom dataset, contrasting geometric fidelity and temporal consistency across competing frameworks.

Figure 10 details supplementary visual evaluations on the PanopticSports dataset, emphasizing our method's enhanced capability in handling complex motion patterns under real-world capture conditions.

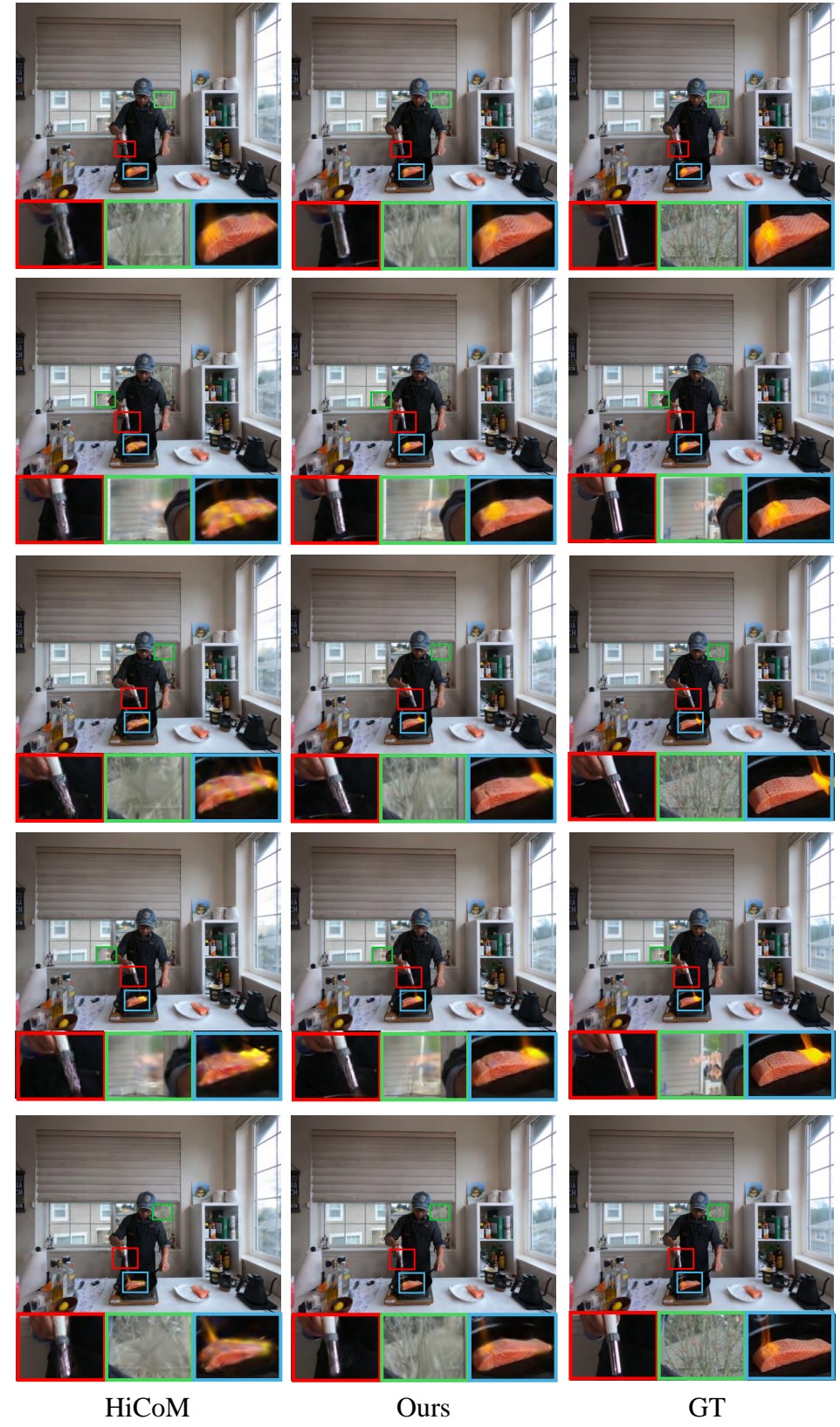

HiCoM                    Ours                    GT

Figure 8: Qualitative results on N3DV Dataset.

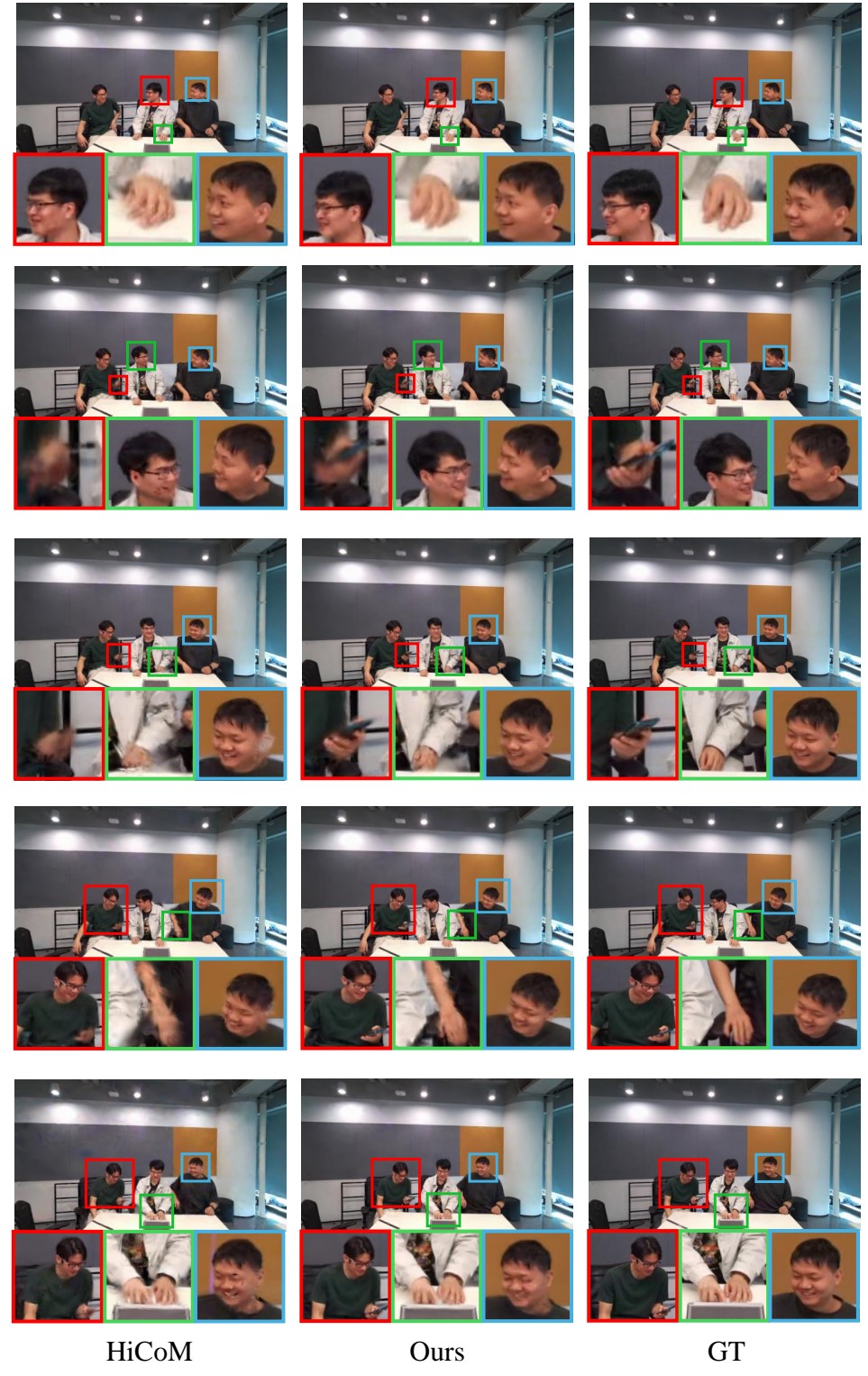

HiCoM                    Ours                     GT

Figure 9: Qualitative results on Meet Room Dataset.

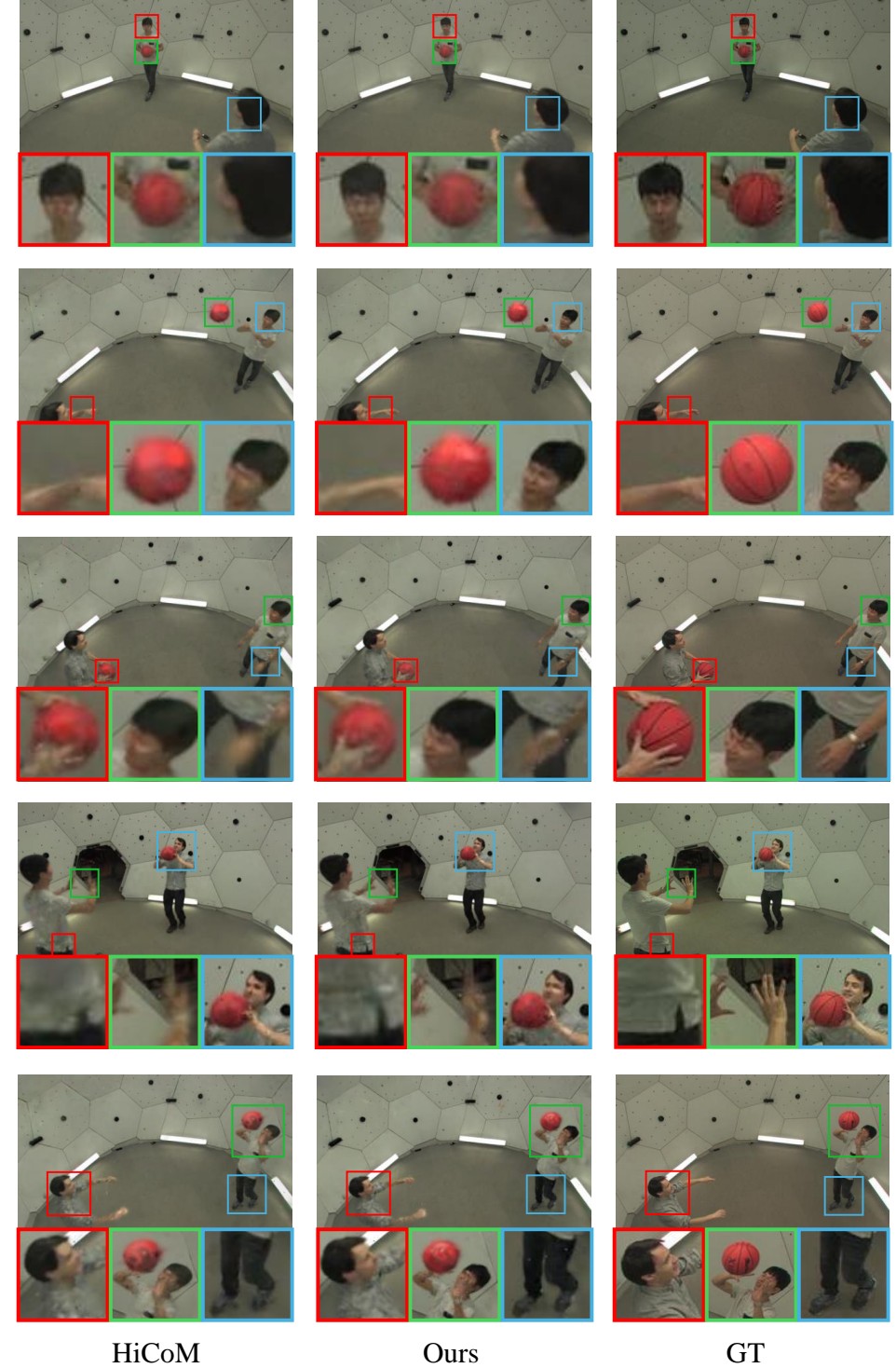

HiCoM            Ours            GT

Figure 10: Qualitative results on PanopticSports Dataset.

