# OpenReview forum: "ReCon-GS: Continuum-Preserved Gaussian Streaming for Fast and Compact Reconstruction of Dynamic Scenes"
_NeurIPS.cc/2025/Conference — NeurIPS 2025 poster_

### Official Review · Reviewer_Yahd · 2025-06-10

**Clarity:** 3
**Significance:** 3
**Originality:** 3
**Rating:** 4
**Confidence:** 3

**Summary:**

This paper proposes ReCon-GS, a method that dynamically assigns multi-level anchor Gaussians to capture inter-frame geometric deformation. This approach reduces storage requirements while maintaining high reconstruction quality. Experiments on multiple datasets shows this method achieves better reconstruction accuracy and storage efficiency than current state-of-the-art online methods.

**Questions:**

- Could the authors visualize the spatial extent and assignment of anchors at each level? This would provide intuitive evidence that the multi-level scheme effectively captures motion in a coarse-to-fine manner.

 - How does the proposed method compare with recent offline baselines such as STG[1] and SplineGS[2]? If ReCon-GS not outperforms these, please consider adding these to Tables 1 and 7. If the proposed online method outperforms these offline methods, a more detailed explanation of why online is better would strengthen the paper. Notably, the results reported for STG and 4DGS on N3DV seem weaker than those in the SplineGS paper, clarification would be helpful.

 - Some tables lack proper units for storage and training time, which may hinder reproducibility and clarity.

 - Would replacing the MLP with an explicit trajectory representation (e.g., as in SplineGS) further improve performance or interpretability?

[1] Spacetime Gaussian Feature Splatting for Real-Time Dynamic View Synthesis. CVPR 2024

[2] SplineGS: Learning Smooth Trajectories in Gaussian Splatting for Dynamic Scene Reconstruction. ICLR 2025

**Ethical Concerns:**

["NO or VERY MINOR ethics concerns only"]

**Final Justification:**

Although NeurIPS restricts the inclusion of visualizations for the hierarchical Gaussian, I trust the authors can elaborate on this aspect thoroughly in the revised appendix. I will maintain my original score.

**Limitations:**

yes

**Quality:**

3

**Strengths And Weaknesses:**

Strengths

 - The idea of coarse-to-fine multi-level anchors is well-motivated and aligns with real-world motion patterns, leading to reduced storage overhead.

 - The experiments are thorough, and the proposed method outperforms existing state-of-the-art online approaches.

Weaknesses

 - The final multi-level anchor modeling lacks visualization, which would help justify its effectiveness and spatial structure.

 - The comparisons in the offline setting omit some recent baselines.

---

> ### Author Rebuttal · Authors · 2025-07-28
>
> We sincerely appreciate your recognition of our Adaptive Hierarchical Motion Representation framework design and particularly value your acknowledgment of the comprehensive experimental validation demonstrating superior performance over existing streaming frameworks. We will now systematically address each of your questions and concerns.
>
> **1. The Visualization of hierarchical Anchor Gaussian & The effectiveness of Anchor Guassian.**
>
> While NeurIPS rebuttal guidelines prevent us from providing visualizations of hierarchical anchor Gaussian, we will include  this visualization in the supplemental appendix of our revised version to demonstrate the efficacy of Anchor Gaussians. Moreover, we have included comprehensive quantitative evidence to prove the effectiveness of Anchor Gaussians in *Table #1*. The data demonstrates that compared to HiCoM[1] (the current open-source SOTA in streaming frameworks), ReCon-GS generates a comparable number of initial Gaussians in the first frame but reduces subsequent incremental Gaussian generation by approximately **60%**. This significant reduction indicates that ReCon-GS achieves high-quality scene reconstruction while requiring only minimal Gaussian increments (approximately 300 per frame). These metrics strongly validate the efficacy of our Adaptive Hierarchical Motion Representation in enabling efficient motion modeling.
>
> **Table #1: Statistics of 3DGS Quantities at Different Stages. (Averaged Over 3 Training Results)**
> | Method         | inital 3DG(↓)| incremental 3DG(↓)| Total 3DG(↓)|
> |----------------|--------------|-------------------|-------------|
> | HiCoM          | 243.3k | 15.8k | 402.1k |
> | **ReCon-GS**   | 244.0k | **6.4k** | **308.8k** |
>
> **2. More Quantitative Results Compared with Recent SOTA Offline Methods.**
>
> Based on your suggestions, we have compared the performance of ReCon-GS with STG[2] and SplineGS[3]. After reviewing their papers, we have also noticed the inconsistency in the benchmarks in the performance comparison between STG and SplineGS: the performance of STG reported by SplineGS (32.57dB) is higher than that in the STG paper (32.05dB). Since SplineGS is not currently open-sourced, we will directly use the performance reported in the paper to ensure fairness.
>
> *Table #2* demonstrates ReCon-GS's **state-of-the-art (SOTA)** performance in rendering quality, storage efficiency, training time, and FPS even when compared to recent offline methods. This superiority stems primarily from the Adaptive Hierarchical Motion Representation's efficient motion modeling capability. Offline methods use global MLPs to represent motion across space and time. This creates compact motion encoding, but storage limits hurt their generalization ability. Swift4D (ICLR25)[4] addresses this problem through static-dynamic 3D gaussian decoupling, allocating more storage to dynamic Gaussian motion representation. However, camera-related issues may introduce noticeable artifacts in static regions[5], compromising static scene reconstruction. Moreover, as training datasets trend toward higher frame rates, streaming frameworks have shifted optimization objectives from temporally global to temporally local deformation field refinement, emerging as a more effective paradigm. Recently, the fundamental challenge in streaming frameworks remains temporal error accumulation. ReCon-GS overcomes this via its novel Dynamic Reconfiguration Strategy, which preserves motion representation fidelity through periodic anchor Gaussian redistribution. This constitutes the underlying reason for ReCon-GS's performance superiority over current SOTA methods.
>
>
> **Table #2: Compare with recent offline baselines.**
> | Category | Method          | PSNR (dB)(↑)| Storage (MB/Frame)(↓) | Training Time (min)(↓) | FPS(↑) |
> |----------|-----------------|-------------|-----------------------|------------------------|--------|
> | Offline  | STG (official)  | 32.05 | 0.67 | - | 140 |
> | Offline  | STG (SplineGS)  | 32.57 | - | 252 | 140 |
> | Offline  | SplineGS        | 32.60 | - | 55  | 76  |
> | Online   | **ReCon-GS(ours)** | **32.66** | **0.44** | **32** | **250** |
>
> **3. Some tables lack proper units for storage and training time.**
>
> We appreciate your suggestion. Providing more detailed units will make our manuscript clearer for readers, and we will add unit information in the tables in the subsequent revised version.
>
> **4. After replacing the explicit motion representation with MLP, can the performance be further improved?**
>
> This is an excellent advice. Indeed, many current online and offline methods use MLP for implicit representation of inter-frame motion. However, for the online framework, implicit representation will bring additional issues:
>  * **The storage of MLP parameters.**
>  Without changing the number of anchor Gaussians, the introduction of MLP will consume more storage compared to the explicit motion representation of ReCon-GS.
>  * **Extremely limited quality improvement.**
> When ReCon-GS needs to sparsify anchor Gaussians due to storage constraints, the MLP-based motion representation method may lead to a huge decline in rendering quality.
>
> Thank you again for your valuable feedback. We hope our responses meet your expectations. If there are any further questions or concerns, please do not hesitate to let us know.
>
> **Reference**:
>
> [1]Qiankun Gao, Jiarui Meng, Chengxiang Wen, Jie Chen, and Jian Zhang. "Hicom: Hierarchical coherent motion for dynamic streamable scenes with 3d gaussian splatting." In Proceedings of the Advances in Neural Information Processing Systems (NeurIPS), 2024.
>
> [2]Zhan Li, Zhang Chen, Zhong Li, and Yi Xu. "Spacetime gaussian feature splatting for real-time dynamic view synthesis." In Proceedings of the IEEE/CVF Conference on Computer Vision and Pattern Recognition (CVPR), 2024.
>
> [3]Jihwan Yoon, Sangbeom Han, Jaeseok Oh, and Minsik Lee. "SplineGS: Learning Smooth Trajectories in Gaussian Splatting for Dynamic Scene Reconstruction." In The Thirteenth International Conference on Learning Representations (ICLR), 2025.
>
> [4]Jiahao Wu, Rui Peng, Zhiyan Wang, Lu Xiao, Luyang Tang, Jinbo Yan, Kaiqiang Xiong, and Ronggang Wang. Swift4D: Adaptive divide-and-conquer Gaussian Splatting for compact and efficient reconstruction of dynamic scene. In International Conference on Learning Representations (ICLR), 2025.
>
> [5]Youngsik Yun, Jeongmin Bae, Hyunseung Son, Seoha Kim, Hahyun Lee, Gun Bang, and Youngjung Uh. "Compensating Spatiotemporally Inconsistent Observations for Online Dynamic 3D Gaussian Splatting." In Proceedings of the Special Interest Group on Computer Graphics and Interactive Techniques Conference Conference Papers (SIGGRAPH), 2025.

---

> > ### Comment · Reviewer_Yahd · 2025-08-01
> >
> > Thanks for the authors' response.  Although NeurIPS restricts the inclusion of visualizations for the hierarchical Gaussian, I trust the authors can elaborate on this aspect thoroughly in the revised appendix.  I will maintain my original score.

---

> > > ### Author Response · Authors · 2025-08-03
> > >
> > > Dear Reviewer Yahd,
> > >
> > > Thank you for your valuable comments and recognition of our work. We will revise our paper accordingly, particularly by incorporating visualizations of Anchor Gaussians to better illustrate their effectiveness for readers.
> > >
> > > Best regards,
> > > The Authors of Paper #5309

---

> > > ### Author Response · Authors · 2025-08-09
> > >
> > > Dear Reviewer Yahd,
> > >
> > > We wish to express our deepest gratitude for your continued engagement throughout the reviewing and discussion phase and your recognition of our work. We fully acknowledge the value of your insights and will diligently incorporate the experiments and visualizations you highlighted into our revised manuscript.
> > >
> > > As the discussion phase approaches its final stage, if there are any further points requiring clarification, we would be honored to provide additional details to the best of our ability. If you find that all issues have been adequately resolved, we respectfully hope that you might consider raising your score.
> > >
> > > Once again, we extend our profound appreciation for the invaluable time, dedication, and constructive feedback you have invested in improving the quality of our research.
> > >
> > > Best regards,
> > >
> > > The Authors of Paper #5309

---

### Official Review · Reviewer_pBHc · 2025-06-18

**Clarity:** 2
**Significance:** 3
**Originality:** 3
**Rating:** 4
**Confidence:** 3

**Summary:**

This paper presents ReCon-GS, a novel framework for real-time free-viewpoint video (FVV) reconstruction that balances high rendering fidelity with memory efficiency. It introduces an Adaptive Hierarchical Motion Representation using Anchor Gaussians to model scene dynamics in a coarse-to-fine manner. A Dynamic Hierarchy Reconfiguration mechanism addresses anchor drift while preserving motion coherence via intra-level deformation inheritance. Additionally, a storage-aware optimization strategy enables flexible trade-offs between quality and memory. Experiments show that ReCon-GS outperforms prior streaming methods with faster training and lower memory usage.

**Questions:**

Most of my questions have been raised in the Weaknesses section, and also is there a explicit motion relation between anchor gaussians across different level?

**Ethical Concerns:**

["NO or VERY MINOR ethics concerns only"]

**Final Justification:**

The authors have addressed my concerns regarding the effectiveness of the Hierarchical Motion Representation, so I have raised my score accordingly.

**Limitations:**

Yes

**Paper Formatting Concerns:**

There is no major formatting issue

**Quality:**

2

**Strengths And Weaknesses:**

Strength:
1. The proposed multi-level anchor-driven deformation model captures scene dynamics in a coarse-to-fine manner, providing an efficient and compact motion representation.
2. The reconfiguration strategy helps mitigate anchor drift and improves temporal coherence.
3. The ablation study and per-scene result is detailed.

Weakness:
1. While the method shows improvement in streaming settings, it underperforms compared to offline methods. In some cases, it requires higher storage, longer training times, and achieves lower reconstruction quality.
2. Some details are not clear to me: how often the dynamic hierarchy reconfiguration is triggered during training and whether the framework is sensitive to this parameter, A clearer formulation explain how Anchor Gaussians quantitatively govern or parameterize general Gaussian deformation. In Table 3: It is surprising that removing the Hierarchical Motion Representation leads to lower storage, which I think is proposed to improve efficiency and compactness.
3. The method relies on freezing appearance attributes initialized from the first frame during the first training phase. This introduces sensitivity to initial conditions when the first frame poorly captures the scene. Do you have examples show that.
I will consider raise rating if my above weakness point are answered.

---

> ### Author Rebuttal · Authors · 2025-07-28
>
> We sincerely appreciate your recognition of ReCon-GS, particularly regarding how its adaptive hierarchical motion representation framework and Dynamic Reconfiguration Strategy effectively achieve efficient and compact motion representation while resolving error accumulation caused by anchor Gaussian drift. Moving forward, we will systematically address each of your concerns and questions.
>
> **1. Regarding storage requirements, rendering quality, and training time compared to offline methods.**
>
> We acknowledge the critical importance of these factors in determining ReCon-GS's practical applicability. It should be noted that through its Storage-aware fidelity optimization objective, ReCon-GS enables storage regulation by adjusting the density of anchor Gaussians in its Adaptive Hierarchical Motion Representation framework while maintaining efficient motion modeling. As evidenced in *Table #1*, ReCon-GS achieves tunable quality-storage trade-offs ranging from 0.14MB/Frame to 0.63MB/Frame across varying anchor Gaussian densities. Crucially, at a 1/64 Anchor Guassian density, ReCon-GS maintains an average storage cost of **0.19MB/Frame** – not only lower than offline methods (4DGS[1], STG[2], Swift4D[3]) but simultaneously achieving **state-of-the-art (SOTA) rendering quality** among all approaches. For training time, online frameworks inherently face disadvantages compared to offline paradigms due to computational constraints, representing an area for our future improvement. However, in real-time 4D reconstruction scenarios, ReCon-GS's online framework consequently offers greater practical utility than offline alternatives.
>
> **Table #1: Quantitative Comparison Across Varying Anchor Gaussian Density Levels. Storage values represent per-frame averages inclusive of the initial frame.**
> |Method      | Anchor Guassian Density  | Storage (MB/Frame)(↓) | PSNR (dB)(↑) |
> |------------|--------------------------|-----------------------|--------------|
> |4DGS        | - | 0.30 | 31.36 |
> |STG         | - | 0.67 | 32.05 |
> |Swift4D     | - | 0.40 | 32.23        |
> |ReCon-GS    | 1/16        | 0.63    | **32.72**|
> |ReCon-GS    | 1/24 (Ours) | 0.44    | 32.66    |
> |ReCon-GS    | 1/32        | 0.37    | 32.50    |
> |ReCon-GS    | 1/48        | 0.28    | 32.39    |
> |ReCon-GS    | 1/64        | 0.19    | 32.31    |
> |ReCon-GS    | 1/96        | **0.14**| 32.11    |
>
> **2. How often the dynamic hierarchy reconfiguration is triggered during training?**
>
> ReCon-GS employs a fixed 4-frame interval for reconfiguration. This design stems from the observation that reconstruction quality inherently fluctuates in streaming frameworks. Consequently, quality-based dynamic adaptation proves ineffective for this issue since quality variations within 3-5 frame intervals do not reliably indicate diminished motion representation capacity of anchor Gaussians. The 4-frame reconfiguration interval represents the optimal configuration determined through extensive experimentation.
>
> **3. How Anchor Gaussians quantitatively govern or parameterize general Gaussian deformation?**
>
> We adopt a simple yet effective paradigm where each General Gaussian directly inherits motion parameters from its associated Anchor Gaussian. Following Dynamic Reconfiguration Strategy, which redistributes Anchor Gaussians, General Gaussians are reassigned to new anchors via L1-distance-based association. After reassignment, General Gaussians subsequently inherit motion transformations from their newly assigned Anchor Gaussians during deformation field training (Stage 1).
>
> **4. Removing the Hierarchical Motion Representation leads to lower storage.**
>
> In *Table #2*, Our storage analysis reveals that removing hierarchical motion representation significantly reduces motion field storage requirements. However, this comes at the cost of diminished motion representation capacity, necessitating increased incremental 3DG generation to compensate for detail loss. Crucially, by implementing spherical harmonics at degree=1 for 3D Gaussians, the per-Gaussian storage footprint remains minimal, preventing substantial memory overhead despite increased Gaussian counts.
>
> **Table #2: The ablation study on storage.**
> |  | Total 3DG number(↓)| Storage for 3DG (MB)(↓) | Storage for Deformation(MB)(↓) | Total Storage(MB)(↓) |
> |----------------------------------------|--------------------|-------------------------|--------------------------------|----------------------|
> | w/o Hierarchical Motion Representation | 320.6k  | 14.68 | 81.05 | 95.73                |
> | Ours (full)   | 308.8k   | 14.13    | 116.12        | 128.25  |
>
> **5. The sensitivity to initial conditions when the first frame poorly captures the scene.**
>
> Your suggestion is profoundly insightful. Streaming frameworks indeed face catastrophic error accumulation when encountering poor quality of initial frame reconstruction. To demonstrate the superiority of ReCon-GS's motion representation framework, we conducted experiments under varying initial frame training iterations.
>
> Table #3 reports ReCon-GS performance under varying initial frame training iterations. At 2,500 iterations, suboptimal initial reconstruction quality and sparse scene representation result in the steepest performance degradation slope. When trained for 5,000 iterations, quality degradation is substantially mitigated. Extending to 10,000 iterations yields converged initial frame training, further optimizing quality decay  while achieving state-of-the-art (SOTA) performance. Responding to your insight, we will integrate this analysis in the future revised version to demonstrate the critical impact of initial frame quality on streaming frameworks while further validating ReCon-GS's superior robustness against error accumulation.
>
> **Table 3: The ablation study on Training Steps of Frame 0.**
> | Method   | Init. Training_Step | PSNR (dB)(↑) | Frame 0 PSNR (dB)(↑)  | Slope (↓) | Storage (MB/Frame)(↓) | Init. 3DG Storage (MB)(↓) | Incr. 3DG Storage (MB)(↓) | Storage for Deformation Field (MB)(↓) |
> |----------|---------------------|--------------|-----------------------|-----------|-----------------------|---------------------------|---------------------------|-----------------------------|
> | ReCon-GS | 2500 | 31.03 | 31.59 | 0.0077 | 0.46 |   8.96 | 42.17 | 78.96|
> | ReCon-GS | 5000 | 32.24 | 32.51 | 0.0029 | 0.48 | 11.14 | 11.57 | 94.80 |
> | ReCon-GS | 10000  | 32.63 | 32.75 | 0.0016  | 0.46  | 11.12 | 4.11 | 92.76 |
>
> **6. Is there a explicit motion relation between anchor gaussians across different level?**
>
> No, We deliberately avoid imposing hierarchical coarse-to-fine constraints on Anchor Gaussians because non-tree-structured organization enhances motion representation efficacy. This design accounts for subtle quasi-rigid motions occurring at rigid body junctions, which would be inadequately represented under tree-structured partitioning. Enforcing tree-structured partitioning would consequently compromise motion encoding efficiency.
>
> We are grateful for your constructive feedback, which has been instrumental in enhancing the quality of our manuscript. Thank you once again for your time and consideration.
>
> **Reference**:
>
> [1]Guanjun Wu, Taoran Yi, Jiemin Fang, Lingxi Xie, Xiaopeng Zhang, Wei Wei, Wenyu Liu, Qi Tian, and Xinggang Wang. 4D Gaussian Splatting for Real-Time Dynamic Scene Rendering. In Proceedings of the IEEE/CVF Conference on Computer Vision and Pattern Recognition (CVPR), 2024.
>
> [2]Zhan Li, Zhang Chen, Zhong Li, and Yi Xu. "Spacetime gaussian feature splatting for real-time dynamic view synthesis." In Proceedings of the IEEE/CVF Conference on Computer Vision and Pattern Recognition (CVPR), 2024.
>
> [3]Jiahao Wu, Rui Peng, Zhiyan Wang, Lu Xiao, Luyang Tang, Jinbo Yan, Kaiqiang Xiong, and Ronggang Wang. Swift4D: Adaptive divide-and-conquer Gaussian Splatting for compact and efficient reconstruction of dynamic scene. In International Conference on Learning Representations (ICLR), 2025.
>
> [4]Qiankun Gao, Jiarui Meng, Chengxiang Wen, Jie Chen, and Jian Zhang. "Hicom: Hierarchical coherent motion for dynamic streamable scenes with 3d gaussian splatting." In Proceedings of the Advances in Neural Information Processing Systems (NeurIPS), 2024.

---

> > ### Comment · Reviewer_pBHc · 2025-08-05
> >
> > Thank you for the authors’ efforts in addressing my concerns. However, I still have questions regarding the validity of the Hierarchical Motion Representation, which appears to be a central contribution of this work.
> >
> > In Table 3 of the main paper, removing Hierarchical Motion Representation leads to a PSNR of 31.43 with a storage of 0.32. Yet in your new ablation table, reducing the motion density (e.g., using 1/96) achieves a lower storage of 0.14 while improving PSNR to 32.1. This seems contradictory and raises questions about the true benefit of the hierarchical design.
> >
> > Given the formulation simplicity of the Anchor Gaussians governing general Gaussian deformation, I remain skeptical about the effectiveness and necessity of the proposed hierarchical structure.
> >
> > Additionally, the paper remains unclear about which Gaussian parameters are optimized during training, particularly, how are the positions of both the general and anchor Gaussians updated?

---

> > > ### Author Response · Authors · 2025-08-06
> > >
> > > Dear Reviewer pBHc,
> > >
> > > We sincerely appreciate your thoughtful comments and the time invested in reviewing our work. We would like to address your concerns as follows:
> > >
> > > **Q1**: "Remaining skeptical about the effectiveness and necessity of the proposed hierarchical structure."
> > >
> > >
> > > **A1**: To help you better understand the two test configurations, we first present in Table 1 the calculation method of anchor Gaussian counts per level in ReCon-GS under two settings: (i) removing the Hierarchical Motion Representation and (ii) Anchor Gaussian Density = 1/96. Then, in Table 2, we report the respective performance of these two configurations. In response to your concerns regarding the effectiveness and necessity of Adaptive Hierarchical Motion Representation framework, we provide the following clarifications in two aspects:
> > >
> > > - **Storage Efficiency**: In terms of Gaussian storage, although the Initial 3D Gaussian storage for ReCon-GS(Anchor Gaussian density=1/96) is comparable to the case of removing the Hierarchical Motion Representation (11.08MB vs. 11.18MB), the incremental 3D Gaussian storage decreases by approximately 15% (3.12MB vs. 3.51MB). This indicates that **sparse yet hierarchical motion representation in ReCon-GS enables more effective reconstruction** compared to the non-hierarchical design, demonstrating **the validity of our Adaptive Hierarchical Motion Representation**. However, it is important to note that the Gaussian storage alone does not represent the full storage cost. For fair ablation, the non-hierarchical version of ReCon-GS must adopt a single-layer anchor density matched to our full ReCon-GS (Anchor Gaussian density=1/24). As shown in Table #2, this results in significantly more anchor Gaussians than our Anchor Gaussian density=1/96 setup (10,173 vs. 3,667 anchors), explaining why the deformation field storage in Anchor Gaussian density=1/96 ReCon-GS **drops by over 60%** compared to the non-hierarchical version. Notably, deformation field storage constitutes ~85% of the total storage. Consequently, Anchor Gaussian density=1/96 ReCon-GS achieves over a 50% reduction in total storage compared to the non-hierarchical design.
> > >
> > > - **The Effectiveness of Adaptive Hierarchical Motion Representation**: In terms of motion representation, **the PSNR improvement** (31.43dB → 32.11dB) provides empirical evidence for the effectiveness of our proposed Adaptive Hierarchical Motion Representation framework. Additionally, **the reduction in incremental Gaussian generation** (~15%) further demonstrates the efficiency of our hierarchical design. While we are unable to share visualizations of the anchor Gaussian distributions for Anchor Density=1/96 due to NeurIPS submission policies, we will include these in the updated version to further validate the effectiveness of our anchor placement and the efficacy of the Adaptive Hierarchical Motion Representation.
> > >
> > > **Table 1. The Configuration on Two Settings. (Total 3DG num = *N*)**
> > > | | Hierarchical Level | Anchor Gaussian Density | Level 1 Anchor | Level 2 Anchor | Level 3 Anchor |
> > > |---|---|---|---|---|---|
> > > | w/o Hierarchical Motion Representation|1|1/24|*N*/24|-|-|
> > > | Anchor Gaussian Density = 1/96|3|1/96|*N*/96|*N*/(96*3)|*N*/(96\*3\*3)|
> > >
> > > **Table 2. The Quantitative Comparison on *w/o Hierarchical Motion Representation* and *Anchor Gaussian Density=1/96*.**
> > >
> > > | | PSNR (dB)(↑) | Avg. Storage (MB/Frame)(↓) | Total Storage (MB)(↓) | Init. 3DG Storage (MB)(↓) | Incr. 3DG Storage (MB)(↓) | Deformation Storage (MB)(↓) | Avg. Level 1 Anchor | Avg. Level 2 Anchor | Avg. Level 3 Anchor |
> > > |---|---|---|---|---|---|---|---|---|---|
> > > | w/o Hierarchical Motion Representation|31.43|0.32|95.92|11.18|3.51|81.23|10173|-|-|
> > > | Anchor Gaussian Density = 1/96|32.11|0.14|43.48|11.08|3.12|29.28|2540|846|281|
> > >
> > > **Q2**: "The paper remains unclear about which Gaussian parameters are optimized during training & How are the positions of both the General and Anchor Gaussians updated?"
> > >
> > > **A2**: Thank you for raising this. We clarify the full training protocol as follows: The first frame (Frame 0) is reconstructed using the standard 3DGS training strategy. For all subsequent frames (i.e., Frame 1 and onwards), our method proceeds in two training stages:
> > >
> > > - **Stage 1 (Explicit Deformation Field Training)**: Only explicit displacement parameters and rotation quaternions parameter of Anchor Gaussians are optimized. The positions of Anchor Gaussians are obtained by combining their explicit displacement parameters with the corresponding 3DG positions, while General Gaussians apply their bound Anchor Gaussian's displacement parameters to their own positions.
> > >
> > > - **Stage 2 (View-based Densification)**: All attributes of incremental Gaussians are optimized. Non-incremental 3D Gaussians are freezed.
> > >
> > >
> > > We deeply value your critical feedback and would be happy to provide further clarifications if needed.
> > >
> > > Best regards,
> > > The Authors of Paper #5309

---

> > > ### Author Response · Authors · 2025-08-07
> > >
> > > Dear Reviewer pBHc,
> > >
> > > Thank you for your time and insightful feedback on our submission. We have carefully addressed all concerns raised in our rebuttal and appreciate your valuable guidance. As the discussion phase nears its end, we kindly ask if our responses have adequately resolved your concerns. We are happy to provide any further details needed. If our responses have adequately addressed your concerns, we would be deeply honored if you were willing to raise your score.
> > >
> > > Thank you again for your thoughtful review.
> > >
> > > Best regards,
> > >
> > > The Authors of Paper #5309

---

> ### Comment · Area_Chair_Ymwr · 2025-08-03
> **Authors' Response**
>
> Dear Reviewer pBHc,
>
> The authors have provided addition data in response to your questions. What is your view after seeing this additional information? It would be good if you could actively engage in discussions with the authors during the discussion phase ASAP, which ends on EoA (Aug 6).
>
> Best,
> AC

---

> > ### Comment · Area_Chair_Ymwr · 2025-08-04
> > **Authors' Response**
> >
> > Dear Reviewer pBHc,
> >
> > The authors have provided detailed answers to your questions. Could you please review them and comment on what your thoughts are based on their responses? Please note, acknowledging that you have read the authors' responses along with the other reviews is a mandatory step of the responsible review process for NeurIPS 2025. Your prompt attention in a timely fashion would be greatly appreciated, so the authors have adequate time to discuss with your before the close of the reviewer-author discussion period (AoE Aug 6).
> >
> > Best,
> > AC

---

### Official Review · Reviewer_zioK · 2025-06-25

**Clarity:** 3
**Significance:** 2
**Originality:** 2
**Rating:** 4
**Confidence:** 3

**Summary:**

This paper proposes a multi-level anchor Gaussian design for real-time free-viewpoint video reconstruction. To support the hierarchical structure, the method also decomposes rigid transformations in a hierarchical manner. Additionally, the Gaussian hierarchy is reconfigured during training to better capture motion. The proposed approach achieves improved performance while offering faster training and reduced storage requirements.

**Questions:**

Please kindly refer to the "Weaknesses".

**Ethical Concerns:**

["NO or VERY MINOR ethics concerns only"]

**Final Justification:**

The authors provide a clear explanation of their method and effectively clarify the novelty of their work. I am raising my score.

**Limitations:**

Yes.

**Paper Formatting Concerns:**

The paper is clearly and effectively written.

**Quality:**

3

**Strengths And Weaknesses:**

### Strengths:
- The writing is clear and easy to follow. The methodology figure is well-designed and effectively conveys the main idea, allowing readers to grasp it quickly.
- The method uses less training cost to get better performance.

### Weaknesses:
- The hierarchical idea in this paper is similar with ScaffoldGS [https://arxiv.org/pdf/2312.00109] as well as the level-of-detail concept in OctreeGS [https://arxiv.org/pdf/2403.17898], with the addition of rigid transformations. As it currently stands, the contribution appears to be more of a technical extension. Further clarification is needed to better highlight the novelty of the proposed approach.
- Only Table 1 reports LPIPS metrics. It would be beneficial to include perceptual metrics in other experiments as well to enable a more comprehensive comparison.
- This is a minor question: does the reported training time include the first-frame reconstruction? According to the paper, this step requires 10,000 to 15,000 iterations, so it seems unlikely to be completed within 10 seconds. Additionally, how many Gaussian primitives are generated in total?

---

> ### Author Rebuttal · Authors · 2025-07-28
>
> We appreciate your recognition of the strengths of our work, particularly the achievement of state-of-the-art (SOTA) performance using a straightforward loss function during training, as well as your positive feedback on the quality of our writing. We will systematically address each of your concerns and questions.
>
> **1. The proposed method shares similarity with Scaffold-GS and Octree-GS.**
>
> ReCon-GS demonstrates fundamental distinctions from Scaffold-GS[1] and Octree-GS[2]. First, Scaffold-GS and Octree-GS target static 3D reconstruction, while ReCon-GS addresses dynamic 4D reconstruction, resulting in divergent frameworks and optimization objectives. Moreover, Scaffold-GS embeds Gaussian attributes via uniformly sampled voxel-grid anchors for compact scene representation. Octree-GS achieves multi-resolution compression through Level-Of-Detail (LOD) hierarchies. Both prioritize spatial compactness. In contrast, ReCon-GS employs adaptively sampled Anchor Gaussians that capture local motion characteristics. Its hierarchical structure compactly encodes inter-frame motion rather than spatial features. Consequently, ReCon-GS’s Adaptive Hierarchical Motion Representation fundamentally differs from these static approaches.
>
> **2. Include perceptual metrics in other experiments.**
>
> Your concern is indeed well-founded, as LPIPS provides a more comprehensive assessment of ReCon-GS's capabilities. As presented in *Table #1*, LPIPS(VGG) metrics comparing ReCon-GS against HiCoM[3] and 3DGStream[4] across both datasets are now included. We observe that ReCon-GS achieves state-of-the-art (SoTA) performance on this perceptual metric among streaming frameworks. Considering your suggestion, we will include the LPIPS metric for the MeetRoom and PanopticSports datasets in our revised manuscript.
>
> **Table #1: The Perceptual Quality Comparison on Meet Room and PanopticSports datasets (Averaged Over 3 Training Results)**
> | Method         | LPIPS at Meet Room (↓) | LPIPS at PanopticSports (↓) |
> |----------------|------------------------|-----------------------------|
> | 3DGStream      | 0.188                  | 0.187                       |
> | HiCoM          | 0.182                  | 0.142                       |
> | **ReCon-GS**   | **0.163**              | **0.136**                   |
>
> **3. Does the reported training time include the first-frame reconstruction?**
>
> We sincerely apologize for not explicitly specifying the methodology in the table caption. The time metrics reported in our tables include the first-frame training duration. Our first-frame training time typically ranges between 300 and 400 seconds, resulting in a negligible impact on the overall reconstruction process. To more clearly demonstrate our method's advantage in low training cost, *Table #2* presents comparisons on the N3DV dataset, showing that ReCon-GS requires significantly less time than both HiCoM and 3DGStream for both first-frame training and average time per frames.
>
> **Table #2: Training Time Comparison on N3DV (Averaged Over 3 Training Results)**
> | Method         | First Frame Tr. Time (sec)(↓) | Avg. Tr. Time (sec)(↓) | Avg. Tr. Time (w/o First Frame)(sec)(↓) |
> |----------------|----------------------------|---------------------|--------------------------------------|
> | 3DGStream      | 341.82                     | 7.58                | 6.46                                 |
> | HiCoM          | 336.51                     | 6.63                | 5.53                                 |
> | **ReCon-GS**   | **328.96**                 | **6.48**            | **5.36**                             |
>
> **4. How many Gaussian primitives are generated in total?**
>
> *Tables #3* and *Table 6* in the manuscript demonstrate that ReCon-GS generates approximately 244k Gaussians in the first frame, which is comparable to HiCoM. However, due to ReCon-GS's implementation of spherical harmonics (SH) at degree=1, its actual first-frame storage occupies only 20% of HiCoM's requirement. During subsequent frame reconstruction, ReCon-GS produces approximately 6.4k incremental Gaussians, merely **40%** of HiCoM's corresponding count. This evidences ReCon-GS's superior motion representation capability, enabling high-quality scene reconstruction through densification while requiring significantly fewer Gaussians.
>
> **Table #3: Statistics of 3DGS Quantities at Different Stages (Averaged Over 3 Training Results)**
> | Method         | inital 3DG(↓)| incremental 3DG(↓)| Total 3DG(↓)|
> |----------------|--------------|-------------------|-------------|
> | HiCoM          | 243.3k       | 15.8k             | 402.1k      |
> | **ReCon-GS**   | 244.0k       | **6.4k**          | **308.8k**  |
>
> Thank you once again for your valuable feedback and insightful comments. We hope that our responses have addressed your concerns and clarified the points raised in your review.
>
> **Reference**:
>
> [1]Tao Lu, Mulin Yu, Linning Xu, Yuanbo Xiangli, Limin Wang, Dahua Lin, and Bo Dai. "Scaffold-gs: Structured 3d gaussians for view-adaptive rendering." In Proceedings of the IEEE/CVF Conference on Computer Vision and Pattern Recognition, 2024.
>
> [2]Kerui Ren, Lihan Jiang, Tao Lu, Mulin Yu, Linning Xu, Zhangkai Ni, and Bo Dai. "Octree-GS: Towards Consistent Real-time Rendering with LOD-Structured 3D Gaussians." IEEE Transactions on Pattern Analysis and Machine Intelligence, 2025.
>
> [3]Qiankun Gao, Jiarui Meng, Chengxiang Wen, Jie Chen, and Jian Zhang. "Hicom: Hierarchical coherent motion for dynamic streamable scenes with 3d gaussian splatting." In Proceedings of the Advances in Neural Information Processing Systems (NeurIPS), 2024.
>
> [4]Jiakai Sun, Han Jiao, Guangyuan Li, Zhanjie Zhang, Lei Zhao, and Wei Xing. 3dgstream: On-the-fly training of 3d gaussians for efficient streaming of photo-realistic free-viewpoint videos. In Proceedings of the IEEE/CVF Conference on Computer Vision and Pattern Recognition (CVPR), 2024.

---

> > ### Comment · Reviewer_zioK · 2025-08-03
> >
> > 1. About similarity between ScaffoldGS and OctreeGS
> >     In your hierarchical anchor architecture, are you suggesting that the inter-frame rigid transformations among the Gaussians attached to the same anchor remain constant, rather than being optimized for spatial alignment within a single frame?
> >
> > 2. I have not other questions about training and performance metrics

---

> > > ### Author Response · Authors · 2025-08-03
> > >
> > > Dear Reviewer zioK,
> > >
> > > Thank you for your response. If you're asking whether General Gaussians tied to the same Anchor Gaussian maintain relatively constant positions, we would be deeply honored to clarify this important point:
> > >
> > > Indeed, within any single hierarchical level, General Gaussians associated with the same Anchor Gaussian share identical deformation parameters, thereby maintaining relative positional invariance. However, to preserve distinct motion characteristics across hierarchical representations, our framework enforces fully independent Anchor-to-General Gaussian matching between different levels. Consequently, through ReCon-GS's multi-level motion representation framework, General Gaussians tied to the same Anchor Gaussian at one hierarchical level may exhibit differential displacement when motions integrate across the hierarchy.
> > >
> > > We sincerely appreciate your patience with this explanation. Should any dimension of this response fall short of your expectations, we would be grateful for the opportunity to provide a more detailed exposition at your convenience.
> > >
> > > Best regards,
> > >
> > > The Authors of Paper #5309

---

> > > > ### Comment · Area_Chair_Ymwr · 2025-08-04
> > > > **Response by Authors**
> > > >
> > > > Dear Reviewer zioK,
> > > >
> > > > The authors have responded to your question about the similarity between ScaffoldGS and OctreeGS. How do you view their response?
> > > >
> > > > Best,
> > > > AC

---

> > > > > ### Comment · Reviewer_zioK · 2025-08-04
> > > > >
> > > > > I see. Are you suggesting that you proposed the anchor architecture to handle deformation hierarchically in order to achieve faster and more efficient motion sampling? Meanwhile, you make the Gaussian deformations independent at the same level under the same anchor to achieve more flexible motion transformation?

---

> > > > > > ### Author Response · Authors · 2025-08-05
> > > > > >
> > > > > > Dear Reviewer zioK,
> > > > > >
> > > > > > We deeply appreciate your thoughtful engagement with our work. Below we provide comprehensive responses to your insightful questions:
> > > > > >
> > > > > > **Q1**: "Are you suggesting that you proposed the anchor architecture to handle deformation hierarchically to achieve faster and more efficient motion sampling?"
> > > > > >
> > > > > > **A1**: Precisely. Our Adaptive Hierarchical Motion Representation (Sec. 4.2) implements hierarchical Anchor Gaussians to enable faster and more efficient motion sampling and representation. Unlike uniform voxel-grid anchors commonly used in prior work, this hierarchy **better adapts to the non-uniform geometric distribution** of reconstructed scene.
> > > > > >
> > > > > > **Q2**: "Do you make Gaussian deformations independent at the same level under the same anchor to achieve more flexible motion transformation?"
> > > > > >
> > > > > > **A2**: In ReCon-GS, General Gaussians tied to the same Anchor Gaussian within the same hierarchy level **share identical motion parameters**. However, the attachments between General Gaussians and Anchor Gaussians are **mutually independent** across different hierarchy levels. Consequently, General Gaussians at the same level under the same anchor can be tied to different anchors at other hierarchy levels, leading to **distinct composite motions** through multi-layer motion integration.
> > > > > >
> > > > > > We sincerely hope these clarifications fully resolve your concerns. Should any questions remain, we would be honored to provide additional elaboration. We are profoundly grateful for your expertise and time invested in reviewing our work. If you find our responses satisfactory, we would be deeply grateful if you consider raising the scores.
> > > > > >
> > > > > > Best regards,
> > > > > >
> > > > > > The Authors of Paper #5309

---

> > > > > > > ### Comment · Reviewer_zioK · 2025-08-05
> > > > > > >
> > > > > > > Thank you for your reply. I have no further concerns.

---

> > > > > > > > ### Author Response · Authors · 2025-08-06
> > > > > > > >
> > > > > > > > Dear Reviewer zioK,
> > > > > > > >
> > > > > > > > We sincerely appreciate your engagement and the time invested in this discussion phase. If you have further inquiries, we would be honored to provide clarification to the best of our ability. We note that multiple reviewers have acknowledged the validity of ReCon-GS’s Adaptive Hierarchical Motion Representation framework and the efficacy of its Dynamic Reconfiguration Strategy.
> > > > > > > >
> > > > > > > > If you determine that all concerns have been adequately addressed, we would be deeply honored if you could kindly reconsider your positive evaluation. Thank you again for your devotion to the review.
> > > > > > > >
> > > > > > > > Best regards,
> > > > > > > > The Authors of Paper #5309

---

### Official Review · Reviewer_43Ni · 2025-07-02

**Clarity:** 3
**Significance:** 2
**Originality:** 2
**Rating:** 4
**Confidence:** 4

**Summary:**

The author proposed a hierarchy-reconfigurable variant of HiCoM. To accomplish such capability of reconfiguration, they repropose a new set of anchor point Gaussians per fixed frame span. To further make sure a smooth transition after hierarchy reconfiguration, they derive the new anchor's deformation field with an inheritance from the previous hierarchy setup.

**Questions:**

- not quite sure the meaning of "Continuum" in the title
- since you always have a stage 1 for training the deformation field after a new frame comes, why bother when a reconfiguration occurs? or does such "inheritance" based deformation field initialization effort help the stage 1 training of the deformation field? Honestly it seems to be marginally helpful.
- Densification is per-frame or per-reconfiguration?
- try eval on Technicolor and longer sequence.
- there actually should be some extra overhead for repetitive anchor reconfiguration: record the index of those anchor points at different timestamp requires extra storage. So how your method achieves a more compact representation than HiCoM?

**Ethical Concerns:**

["NO or VERY MINOR ethics concerns only"]

**Final Justification:**

I have kept the original rating of weak accept for this paper. The method is straightforward enough for me and makes sense with no doubt. The reason for this weak decision is that the method is a subtle update of HiCom, and personally for me it is not a significant innovation. Yet it is still a solid paper with reasonable method and convincing evaluation. I will recommend this paper for acceptance.

**Limitations:**

as mentioned above, it is still a deformation-based method, simply a better way (easier-to-train actually) to represent the gaussian deformation across frames. Just like any other deformation-based dynamic modeling, it cannot efficiently encode highly dynamic scenes.

**Paper Formatting Concerns:**

Null

**Quality:**

3

**Strengths And Weaknesses:**

Good:
- it works great with best performance compared with other sota works. Also baseline works are appropriately selected.
- I do like Figure 6 in appendix which demonstrate the effectiveness of the reconfiguration and would encourage the author to put it in the front sections. Also it will be more convincing to eval the proposed method on longer dataset than those 300s datasets.

Weakness:
- grid based anchor point sampling is BORING.
- **the research innovation of the work is very limited.** The main difference compared with previous HiCom is merely this reconfiguration setup, which is implemented in a naive method: repropose a new set of anchors. The inheritance seems to be unnecessary.
- still it is a deformation-based method. Just like any other deformation-based dynamic modeling, it cannot efficiently encode highly dynamic scenes.

others:
- overall its a well-written paper with abundant visualization. yet readers with little background for HiCom might get lost.

---

> ### Author Rebuttal · Authors · 2025-07-30
>
> We sincerely appreciate your recognition of ReCon-GS's performance. We will now systematically address the questions you have raised.
>
> **1. The meaning of "Continuum" in the title.**
>
> The term "Continuum" highlight ReCon-GS's characteristics. Through our Adaptive Hierarchical Motion Representation and Dynamic Reconfiguration Strategy, the framework effectively prevents progressive geometric distortion in streaming reconstruction – a capability conclusively demonstrated in Figure 7. This preservation of geometric consistency across frames fundamentally realizes a true continuum of scene representation, justifying the terminology.
>
> **2. How essential is Intra-Hierarchical Deformation Inheritance to achieving performance gains during Re-Hierarchization, considering deformation fields undergo per-frame retraining in Stage 1?**
>
> Given that inter-frame motion typically exhibits inertia, ReCon-GS initializes the motion state of the current frame’s Anchor Gaussians using that of the previous frame’s Anchor Gaussians. However, sustained displacements reduce Anchor Gaussians' capacity to represent quasi-rigid motion. This necessitates reassigning anchors through Re-Hierarchization. After reassignment, directly inheriting motion from old to new Anchor Gaussians would compromise temporal motion consistency. To address this, as detailed in Section 4.3, we proposes Intra-Hierarchical Deformation Inheritance. This mechanism enables new anchors to precisely inherit local motion inertia by uniformly acquiring deformation traits from three legacy anchors, which enables local motion inertia inheritance for new anchors while preserving temporal consistency.
>
> To thoroughly validate the efficacy of this mechanism, we conducted an ablation study. As shown in *Table #1*, removing Intra-hierarchical Deformation Inheritance leads to a performance deterioration, accompanied by a pronounced increase in the rate of performance decline. This result confirms the critical role of this mechanism in maintaining motion prediction accuracy and temporal consistency. To enhance readers' understanding of ReCon-GS's validity, we will add this ablation experiment in the revised version based on your feedback.
>
> **Table #1: Ablation on Intra-hierarchical Deformation Inheritance.**
> |  | PSNR (dB)(↑) | Slope (↓) |
> |----------------------------------------|--------------------|-------------------------|
> | w/o Intra-hierarchical Deforamtion Inheritance | 32.45  |  0.0014   |
> | Ours (full)   | 32.66   | 0.0005  |
>
> **3. Densification is per-frame or per-reconfiguration?**
>
> Densification operates on a per-frame scheme. Although this incurs a modest training cost, the frame-by-frame densification promptly compensates for limitations in ReCon-GS's quasi-rigid motion representation regarding local detail reconstruction.
>
>
> **4. Quantitative Result Comparison on other Datasets.**
>
> Based on your suggestion, we conducted supplementary quantitative evaluation on the Technicolor dataset[1].
>
> As demonstrated in *Table #2*, our method achieves **state-of-the-art (SOTA)** performance across rendering quality, storage efficiency, and rendering speed metrics. Notably, compared to streaming approaches, ReCon-GS delivers approximately **0.5 dB higher rendering quality** while reducing storage by over **40%** and significantly accelerating rendering speed. Against offline methods, ReCon-GS attains SOTA rendering quality while achieving more than 25% storage reduction.
>
> **Table #2: Quantitative Comparison on Technicolor Light Field dataset.**
> | Category   | Method         | PSNR (dB)(↑) | SSIM (↑) | Storage (MB/Frame)(↓) | FPS (↑) |
> |------------|----------------|--------------|----------|-----------------------|---------|
> | NeRF-based | HyperReel[2]   | 31.80        | 0.906    | 1.20                  | 4       |
> | Offline    | STG[3]         | 33.60        | -        | 1.10                  | 87      |
> | Offline    | Ex4DGS[4]      | 33.62        | 0.916    | 2.81                  | 72      |
> | Online     | E-D3DGS[5]     | 33.24        | 0.907    | 1.54                  | 79      |
> | **Online** | **ReCon-GS**   | **33.83**    | **0.932**| **0.81**              | **207** |
>
>
>
> **5. How our method achieves more compact deformation data representation than HiCoM?**
>
> Unlike HiCoM, our method doesn't require indexing every deformation field parameter for preservation. ReCon-GS effectively achieve this through a geometrically ordered storage scheme: Our anchor Gaussians, sampled via grid-based Farthest Point Sampling (FPS), undergo motion data serialization relative to the world coordinate, where parameters are stored sequentially from the farthest anchor point from the origin inward along a predetermined spatial axis. This coordinate-based ordering inherently eliminates redundant index storage while ensuring consistent data retrieval, making ReCon-GS significantly more compact than HiCoM.
>
> Thank you once again for your time and effort in reviewing our manuscript. We look forward to any further comments you may have.
>
> **Reference**:
>
> [1]Neus Sabater, Guillaume Boisson, Benoit Vandame, et al. Dataset and pipeline for multi-view light-field video. In Proceedings of the IEEE/CVF conference on Computer Vision and Pattern Recognition Workshops (CVPR workshops), 2017.
>
> [2]Benjamin Attal, Jia-Bin Huang, Christian Richardt, Michael Zollhoefer, Johannes Kopf, Matthew O’Toole, and Changil Kim. HyperReel: High-fidelity 6-DoF video with ray-conditioned sampling. In Proceedings of the IEEE/CVF conference on Computer Vision and Pattern Recognition (CVPR), 2023.
>
> [3]Zhan Li, Zhang Chen, Zhong Li, and Yi Xu. "Spacetime gaussian feature splatting for real-time dynamic view synthesis." In Proceedings of the IEEE/CVF Conference on Computer Vision and Pattern Recognition (CVPR), 2024.
>
> [4]Junoh Lee, ChangYeon Won, Hyunjun Jung, Inhwan Bae, and Hae-Gon Jeon. "Fully explicit dynamic gaussian splatting." In Proceedings of the Advances in Neural Information Processing Systems (NeurIPS), 2024.
>
> [5]Jeongmin Bae, Seoha Kim, Youngsik Yun, Hahyun Lee, Gun Bang, and Youngjung Uh. "Per-gaussian embedding-based deformation for deformable 3d gaussian splatting." In Proceedings of the European Conference on Computer Vision (ECCV), 2024.

---

> > ### Comment · Reviewer_43Ni · 2025-08-04
> >
> > If indexing is also a key difference between your proposal and HiCom, i would suggest you to include it in your paper explicitly also. I dont have other comments considering the method is straightforward enough and makes sense for me, as i have mentioned in my original review. Generally it is a solid paper, and I will keep my weak ac decision.

---

> > > ### Author Response · Authors · 2025-08-04
> > >
> > > Dear Reviewer 43Ni,
> > >
> > > Thank you for your valuable feedback and recognition of our work. Based on your suggestion, we will incorporate a detailed description of key differences in motion parameterization between ReCon-GS and HiCoM, along with quantitative comparisons on the Technicolor dataset in our revised manuscript.
> > >
> > > Best regards,
> > >
> > > The Authors of Paper #5309

---

> > > > ### Comment · Reviewer_43Ni · 2025-08-05
> > > >
> > > > You can borrow some idea from HAC who visualizes its difference from scaffoldGS in its teaser.

---

> > > > > ### Author Response · Authors · 2025-08-06
> > > > >
> > > > > Dear Reviewer 43Ni,
> > > > >
> > > > > We are deeply grateful for your constructive suggestion. Upon reviewing the HAC's visualization in light of your feedback, we fully acknowledge its potential to more clearly highlight the distinctions between our approach and HiCoM. We sincerely appreciate your expertise and will prioritize incorporating this enhancement into our revised manuscript.
> > > > >
> > > > > If you have any further questions or require additional clarification, we would be honored to assist you to the best of our ability. If our responses have adequately addressed your concerns, we would be deeply honored if you were willing to raise your score.
> > > > >
> > > > > Thank you once again for your invaluable time, thoughtful insights, and dedication to improving the quality of our work.
> > > > >
> > > > > Best regards,
> > > > >
> > > > > The Authors of Paper #5309

---

> ### Comment · Area_Chair_Ymwr · 2025-08-03
> **Authors' Response**
>
> Dear Reviewer 43Ni,
>
> The authors have provided addition data in response to your questions. What is your view after seeing this additional information? It would be good if you could actively engage in discussions with the authors during the discussion phase ASAP, which ends on EoA (Aug 6).
>
> Best,
> AC

---

### Comment · Area_Chair_Ymwr · 2025-08-02
**Discussion with Authors**

Dear Reviewers,

The discussion period with the authors has now started. It will last until Aug 6th AoE. The authors have provided responses to your questions.

For those of you, who have already started discussions with the authors, thank you! Others, I request that you please read the authors' responses, acknowledge that you have read them and start discussions with the authors RIGHT AWAY if you have further questions, to ensure that the authors enough time to respond to you during the discussion period.

Best,
AC

---

### Note · Authors · 2025-08-14

Dear Area Chairs and Reviewers,

We extend our deepest gratitude to all chairs and reviewers for your invaluable time and meticulous feedback. We sincerely appreciate the reviewers' consensus that ReCon-GS achieves *state-of-the-art (SOTA) performance in streaming methods*. The Adaptive Hierarchical Motion Representation (AHMR) framework is praised for *its innovative, compact design*, and the Dynamic Reconfiguration Strategy effectively *addresses error accumulation problem*. All reviewers praised *the paper's well-structured presentation and comprehensive experiments*, including baselines and ablations, which *validate ReCon-GS’s superior performance and robustness*.

With the discussion phase concluded, three reviewers (43Ni, zioK, Yahd) explicitly acknowledged that *our revisions have adequately addressed their concerns*. Reviewer pBHc raised a new issue regarding the effectiveness of the proposed hierarchical structure, and we have provided further evidence to validate the effectiveness of the AHMR framework. Next, we will clarify the modifications to the revised manuscript and provide additional clarification to address reviewer pBHc's specific concern.

- **Reviewer 43Ni:** Following your suggestion, we will incorporate visualization diagrams inspired by HAC’s teaser figure to better illustrate ReCon-GS’s innovations and enhance readers’ understanding of its novelty.

- **Reviewer zioK:** We will add the missing perceptual quality metrics in two dataset evaluations in the revised version and further clarify the per-frame training time and 3D Gaussian counts of ReCon-GS. The relationship between Anchor and General Gaussians will be elaborated in future version.

- **Reviewer pBHc:** Regarding your concerns about the AHMR framework, we provide detailed experimental configurations and performance comparisons to confirm the effectiveness and necessity of the proposed hierarchical structure. We included ReCon-GS evaluations under varying Anchor Gaussian densities in the camera-ready version to highlight its SOTA performance in both reconstruction and storage efficiency.

- **Reviewer Yahd:** We will add missing unit annotations in tables and include anchor Gaussian visualizations to enhance readers’ understanding.

We believe these clarifications and supplementary experiments will better align ReCon-GS with the rigorous standards of NeurIPS 2025, and we sincerely appreciate your continued attention to our work.

Best regards,
The Authors of Paper #5309

---

### Decision · Program_Chairs · 2025-09-17

**Decision:**

Accept (poster)

**Comment:**

This paper proposes an adpative storage-efficient method for 4D Gaussian reconstruction of dynamic scenes in a streaming fashion. Its core idea is to employ a hierarchical motion representation to compactly encode inter-frame Gaussian differences and a reconfiguration strategy for them to reduce their drift over time. The paper shows quantitatively superior results over the state of the art. Four reviewers provided final ratings of 4 x borderline accept. The strength of the work was noted in having strong quantitative performance and a logical and simple method. During the author-reviewer discussion phase, most of the reviewer's major concerns were sufficiently addressed and they all raised their scores towards a favorable accept rating. The AC has carefully checked the paper and its reviews, concurs with the reviewers' consensus and recommends acceptance. Congratulations! The authors should incorporate the changes that they have promised into the final camera-ready version of their manuscript.